



# Origins of Multi-decadal Variability in Sudden Stratospheric Warmings

Oscar Dimdore-Miles[1], Lesley Gray[1,2], and Scott Osprey[1,2]

[1]Atmospheric, Oceanic and Planetary Physics, Department of Physics, University of Oxford, OX1 3PU, UK
[2]National Centre for Atmospheric Science, Oxford, OX1 3PU

**Correspondence:** Oscar Dimdore-Miles (oscar.dimdore-miles@physics.ox.ac.uk)

**Abstract.** Sudden Stratospheric Warmings (SSWs) are major disruptions of the Northern Hemisphere (NH) stratospheric polar vortex and occur on average approximately 6 times per decade in observation based records. However, within these records, intervals of significantly higher and lower SSW rates are observed suggesting the possibility of low frequency variations in event occurrence. A better understanding of factors that influence this decadal variability may help to improve predictability

of NH mid-latitude surface climate, through stratosphere-troposphere coupling. In this work, multi-decadal variability of SSW events is examined in a 1000-yr pre-industrial simulation of a coupled Atmosphere-Ocean-Land-Sea ice model. Using a wavelet spectral decomposition method, we show that hiatus events (intervals of a decade or more with no SSWs) and consecutive SSW events (extended intervals with at least one SSW in each year) vary on multi-decadal timescales of period between 60 and 90 years. Signals on these timescales are present for approximately 450 years of the simulation. We investigate the possible

source of these long-term signals and find that the direct impact of variability in tropical sea surface temperatures, as well as the associated Aleutian Low, can account for only a small portion of the SSW variability. Instead, the major influence on long-term SSW variability is associated with long-term variability in amplitude of the stratospheric quasi biennial oscillation (QBO). The QBO influence is consistent with the well known Holton-Tan relationship, with SSW hiatus intervals associated with extended periods of particularly strong, deep QBO westerly phases. The results support recent studies that have highlighted the role of

vertical coherence in the QBO when considering coupling between the QBO, the polar vortex and tropospheric circulation.





## 1   Introduction

Major Sudden Stratospheric Sudden Warming (SSW) events involve significant disruption of the Northern Hemisphere (NH) stratospheric polar vortex and represent the largest mode of interannual variability in the boreal winter stratosphere (Butler et al., 2017; Baldwin et al., 2020). They are associated with an equatorward shift and deceleration of the North Atlantic jet stream (Kidston et al., 2015), negative phases of the North Atlantic Oscillation (NAO) (Baldwin and Dunkerton, 2001) as well as cold snaps over Eurasia and North America (Kretschmer et al., 2018). In reanalysis datasets, SSWs occur at an average rate of 0.6 events/winter but this varies markedly over the record (Butler et al., 2015) suggesting the possibility of variability on much longer timescales. For example, observational studies have noted a hiatus in the 1990s when very few major SSW events occurred (Butler et al., 2015; Pawson and Naujokat, 1999; Shindell et al., 1999) while, in contrast, the early 21st century displayed a remarkable number of consecutive winters containing SSW events (Manney et al., 2005) as well as an anomalously warm stratospheric NH winter polar cap.

Despite a significant body of work aimed at understanding the nature of SSWs and their impacts on mid-latitude surface climate, variability of their occurrence on decadal to multi-decadal timescales is not well understood. Multi-decadal stratospheric variability has been considered in the context of forced anthropogenic warming signals in GCMs. For example Garfinkel et al. (2017) analyse decadal-scale variations in polar vortex strength in a set of historical simulations and propose that an observed hiatus in Eurasian surface warming was most likely due to variability in midwinter vortex strength. Whether the vortex variability was forced by greenhouse gas concentrations or arose through internal variability was not established. Schimanke et al. (2011) find similar multidecadal scale variations of the polar vortex (specifically SSWs) with periods of approximately 52 years in a multi-century GCM integration and propose coherent variability in other parts of the climate system, including vertically propagating planetary wave activity, Eurasian snow cover and Atlantic sea surface temperatures (SSTs). Despite providing some indications of externally driven variability, results from this study are not conclusive, since the GCM used (EGMAM: ECHO-G with Middle Atmosphere Model) exhibits significant bias in mean SSW rate compared to reanalyses (2 events per decade) which means that their findings may not be fully representative of the observed stratosphere. The authors note that further simulations are required to understand this variability. Manzini et al. (2012) explore causes of 20 year period variability in a simulation with prescribed, pre-industrial SSTs. They propose that, given the boundary conditions in the simulations are fixed, such variability must be internally generated. Butchart et al. (2000) suggest that decadal variability in vortex strength as well as SSW frequency may originate from feedbacks caused by the non linear nature of boreal winter stratospheric dynamics. Both works show that these internally induced signals significantly influence mid-latitude surface variability, forcing similar period signals in the NAO and north Atlantic SSTs.

A region often considered in studies of vortex strength variability is the equatorial stratosphere. The primary mechanism for coupling between these regions is between the Quasi Biennial Oscillation (QBO) and the vortex. An association between the phase of the QBO and the strength of the polar vortex was first proposed by Holton and Tan (1980) and Holton and Tan (1982) who found that the polar vortex exhibited a strengthening when the QBO near the 50 hPa level was in its westerly phase (QBO-W) compared to its easterly phase (QBO-E). This link, usually referred to as the Holton-Tan (HT) link, has been reported





in subsequent studies with more comprehensive observations as well as in modelling studies using GCMs including the Met Office Hadley Centre Model 2 (HadGEM2), the predecessor of the model considered in this study (Baldwin and Dunkerton, 1991; Pascoe et al., 2005; Lu et al., 2008; Watson and Gray, 2014). A number of physical mechanisms have been proposed to account for the observed coupling between the QBO and the vortex that involve a QBO influence on wave propagation into the winter stratosphere (Baldwin et al., 2001).

The QBO is typically defined by the equatorial zonal-mean zonal wind (ZMZW) at a single level in the mid-stratosphere. The 50 hPa level is usually used for NH observational studies (Baldwin et al., 2001; Baldwin and Dunkerton, 1998) but some studies have also noted the importance of characterising the vertical structure of the QBO (Fraedrich et al., 1993; Wallace et al., 1993; Baldwin and Dunkerton, 1998; Dunkerton, 2017; Gray et al., 2018; Andrews et al., 2019). In an observational-based study Gray et al. (2018) find an enhanced association between the QBO and polar vortex when a metric incorporating

the vertical coherence of equatorial winds via empirical orthogonal functions is utilised (Schenzinger, 2016). In a model-based study Andrews et al. (2019) introduce a similar but simpler methodology by defining the QBO as the average ZMZW between two vertical levels, which preferentially selects time intervals that display a vertically coherent QBO phase between the specified levels. These studies suggest the importance of vertical QBO metrics when considering QBO-vortex coupling although the influence mechanisms are not well understood.

Decadal to multi-decadal scale variability in the QBO and the HT relationship has also been examined. There are clear variations in QBO period and phase transition timing (Pascoe et al., 2005; Anstey and Shepherd, 2008; Yang and Yu, 2016). These may be linked to variations in the degree of 'stalling' of the QBO phase descent, which can cause more or less persistent wind direction at a given level. A number of studies have also noted the transient nature of the strength of the HT relationship (Lu et al., 2008, 2014; Anstey and Shepherd, 2008; Osprey et al., 2010). Lu et al. (2008, 2014) note that the mid-latitude

wave-guide is modulated by the shape of the vortex so that planetary waves are diverted further equatorwards when the vortex is anomalously strong and wide, and this could temporarily reduce the influence of the QBO on the vortex.

Vortex variability has also been closely associated with variations in winter surface climate that can determine the strength of mid-latitude tropospheric wave driving. Among the most notable of these is the climatological low pressure system over the Aleutian Islands in the Bering sea - the Aleutian low (AL). The depth of the AL has been shown to modulate vertical planetary

wave propagation into the vortex region via coupling with the corresponding stratospheric high on the vortex edge (Woo et al., 2015). The effect has been found in reanalysis (Hu and Guan, 2018) as well as modelling studies (Kren et al., 2016; Kang and Tziperman, 2017). The AL is a key indicator of Pacific climate variability with teleconnections with both tropical and mid-latitude climate (Nitta and Yamada, 1989; Trenberth and Hurrell, 1994; Zhang et al., 1997) and it varies significantly on decadal to multi-decadal timescales. Overland et al. (1999) note that 10 year mean values of SLP over the AL region exhibit

fluctuations of up to 35% of the climatological mean. Subsequent studies corroborate the presence of these decadal scale fluctuations: Sugimoto and Hanawa (2009) and Minobe (1999) show 20 year fluctuations in intensity and centre of action of the AL while Raible et al. (2005) propose a 50-60 year trend in AL depth, suggesting the existence of even longer timescale variability.





Further surface features linked to vortex variability involve tropical SSTs. For example the SST anomalies over the Eastern

Pacific region associated with the El Niño Southern Oscillation (ENSO) have been shown to induce a stratospheric vortex circulation response via a pathway including the AL (Domeisen et al., 2019). A positive ENSO phase is associated with a deepening of the AL which promotes stronger planetary wave forcing of the middle atmosphere. This teleconnection has been found extensively in observation based studies (Garfinkel and Hartmann, 2008; Ineson and Scaife, 2009; Smith and Kushner, 2012) as well as modelling studies (Bell et al., 2009; Domeisen et al., 2014; Manzini et al., 2006; Richter et al.,

2015). However, the connection's robustness has also been shown to vary between ENSO events (Deser et al., 2017; Iza et al., 2016). Other Tropical regions also exhibit elements of connection to vortex strength variability: Rao and Ren (2017) show that Tropical Atlantic SSTs appear to give rise to a vortex response although the response is highly variable throughout the season while Rao and Ren (2015) also propose a Tropical Indian Ocean (TIO) connection which manifests as a competing mechanism with the ENSO-vortex effect, i.e. positive TIO SST anomalies lead to a reduced strength of the AL that weakens the Rossby

wave forcing of the middle atmosphere.

Although substantial effort has been applied to characterising SSWs and their underlying mechanisms, there is little understanding of periods of hiatus (such as that in the 1990s) and consecutive-event years (early-mid 2000s), primarily because of the short record of reliable observations as well as the complexities of the multiple observed teleconnections. Much longer time-series are required to successfully identify and understand the source of decadal and multi-decadal scale variability. In

the meantime, analysis of variability and teleconnections in long climate model simulations may help to understand these processes.

In this work, we analyse long-term variability of the stratospheric polar vortex in a 1000-yr pre-industrial (piControl) simulation of the UK Earth System Model (UKESM). The absence of external forcings such as greenhouse gas increases, volcanic and solar variations allows us to examine sources of long-term variability that are internally generated within the climate system.

We identify intervals containing high and low SSW rates and analyse their variability, with a focus on multi-decadal scales. Improved understanding and representation of stratospheric variability will help to improve predictions of NH winter surface weather and climate (Kidston et al., 2015; Gray et al., 2020). A variety of techniques including wavelet spectral decomposition and cross-spectrum analysis are employed to examine associations with those parts of the climate system that are known to exhibit long-term memory, namely the tropical SSTs and the related Aleutian Low. In addition, we investigate interactions

between the polar vortex and the QBO as a potential source of internally-driven variability. The latter reveals an unexpected source of multi-decadal scale variability associated with the strength and vertical depth of the QBO. The paper is structured as follows: Section 2 sets out the GCM used in the investigation, the spectral analysis method (wavelet analysis) and relevant climate indices. Section 3 presents results from the analysis. Section 4 discusses findings and concludes.





## 2 Model and Data

### 2.1 Model Configuration

The first version of the UK Earth System Model (henceforth referred to as UKESM) is the most recent configuration of the MetOffice unified model (the UM) (Mulcahy et al., 2018). UKESM is a stratosphere resolving coupled ocean-atmosphere-land-sea ice model. The Atmospheric component is GA7.1 which contains 85 vertical atmospheric levels with a maximum altitude of 85km and 35 levels above a height of 18km (Walters et al., 2019; Williams et al., 2018). For this study GA7.1 is run at N96 horizontal resolution (approximately 135 km near the equator). The ocean model used is GO6.0 (Storkey et al., 2018) which contains 75 levels and runs at 1° horizontal resolution. Land surface and sea-ice processes are represented by JULES (GL7.0, Walters et al., 2019) and CICE (GSI8.1, Ridley et al., 2018) models respectively, while ocean biochemistry is added through MEDUSA (Yool et al., 2013). UKESM also includes a fully interactive chemistry scheme via coupling with the UK Chemistry and Aerosols model (UKCA, Mulcahy et al., 2018). We utilise a 1000 year pre-industrial (PI) control simulation of UKESM submitted to CMIP6 which is spun-up to achieve initial model equilibrium following the method outlined in Yool et al. (2020). This run is forced using CMIP6 pre-industrial values for concentrations of major GHGs (global mean 284.317ppm $CO_2$, 808.25ppb $CH_4$, 273.02ppb $N_2O$). While there are no volcanic eruptions in the simulation, background stratospheric volcanic aerosols are set to climatological values between 1850 and 2014 estimated from satellite products and other model simulations (Menary et al., 2018). We choose a PI control for this analysis to examine internal variability in SSWs on multi-decadal timescales. For comparison between the model and the real atmosphere we utilise the ERA-Interim dataset described fully in Dee et al. (2011).

### 2.2 Wavelet Analysis

In order to study possible multi-decadal variability in SSW occurrence, we utilise a wavelet analysis method based on Torrence and Compo (1998). Such a wavelet analysis can be used to examine time series which displays non-stationary spectral power over multiple frequencies (Daubechies, 1990). The wavelet transform of a uniform 1-dimensional time series, $x$, of length $N$ and timestep $\delta t$ is given by the convolution between the series and a scaled and translated version of a wavelet function $\psi_0$ (equation 1)

$$W_n(s) = \sum_{n'=0}^{N-1} x_{n'} \psi^* \left[ (n'-n) \frac{\delta t}{s} \right], \tag{1}$$

where $*$ denotes the complex conjugate and $s$ is the wavelet scale indicating the frequency of the wavelet. Varying $s$ and translating along the time scale (the index $n$), $W_n$ indicates the amplitude of signals at different scales and their variation in time. Torrence and Compo (1998) suggest an approach to varying the scale s as increasing in powers of 2 according to

$$s_j = s_0 2^j, j = 0, 1, ..., J \tag{2}$$





$$J = \delta j^{-1} log_2 \left( \frac{N\delta t}{s_0} \right), \tag{3}$$

where $s_0$ is the shortest resolvable scale of a signal, J corresponds to the longest and $\delta j$ is the scale resolution. The translated
and scaled wavelet has the form

$$\psi^* \left[ (n'-n)\frac{\delta t}{s} \right] = \left( \frac{\delta t}{s} \right)^{1/2} \psi_0 \left[ (n'-n)\frac{\delta t}{s} \right] \tag{4}$$

and we select the form of $\psi_0$ following the recommendation of Torrence and Compo (1998) as a Morlet wavelet, an oscilla-
tory function enveloped by a Gaussian which is expressed as

$$\psi_0(p) = \pi^{-1/4} e^{i\omega_0 p} e^{\frac{p^2}{2}}. \tag{5}$$

The advantages of using a Morlet wavelet for analysing signals in climate time-series is discussed in Lau and Weng (1995)
in which the authors acknowledge that while truly physical signals should be detected regardless of wavelet basis chosen,
for best results one should adopt a wavelet function reminiscent of the real signal. They show that when a Morlet wavelet
form is utilised, spectral decomposition methods can detect common forms of behaviour exhibited in the variability of time
series associated with the Earth's climate. These include time variations in period and amplitude of signals, abrupt changes in
periodicity (sudden regime shift to different spectral behaviour) and some forms of rapid changes in series over time. These
forms of behaviour are most likely relevant for our analysis of SSWs therefore we proceed with a wavelet of this form.

It is computationally quicker to compute the wavelet transform in discrete Fourier space. By the convolution theorem, the
transform reduces to multiplication

$$W_n(s) = \sum_{k=0}^{N-1} \hat{x}_k \hat{\psi}^*(s\omega_k) e^{i\omega_k n\delta t}, \tag{6}$$

where $\hat{x}_k$ and $\hat{\psi}$ are the discrete Fourier transforms of the time series $x$ (equation 7) and the wavelet function (equation 8)
respectively,

$$\hat{x}_k = \frac{1}{N} \sum_{n=0}^{N-1} x_n e^{\frac{-2\pi ikn}{N}} \tag{7}$$

$$\hat{\psi}(s\omega_k) = \left( \frac{2\pi s}{\delta t} \right) \pi^{-1/4} H(\omega_k) e^{-(s\omega_k - \omega_0)^2/2}. \tag{8}$$

$H(\omega_k)$ is the Heaviside function and $\hat{\psi}$ is normalised to have unit energy when integrated over all $\omega$. The square modulus
of the wavelet transform gives the wavelet power spectrum which indicates relative strength of signals in the time series as





a function of signal period and discretised time. We also define a confidence interval for wavelet power observed at a given period and time for a series by assuming a mean background spectrum corresponding to that of a first order autoregressive (AR1, red noise) process modelled by

$$x_n = \alpha x_{n-1} + z_n, \tag{9}$$

where $\alpha$ is the lag-1 autocorrelation of the time series and $z_n$ is Gaussian white noise. Torrence and Compo (1998) show that such a process's wavelet power spectrum is $\chi^2$ distributed and therefore can be used to define a 95% confidence interval for any observed power.

### 2.2.1  Cross Wavelet Spectra

The cross wavelet spectrum of two time series $x$ and $y$ with associated wavelet spectra $W_n^x$ and $W_n^y$ gives a measure of
coincident power (the same period at the same timepoints) between the series. It is given by

$$|W_n^{xy}(s)| = |W_n^{x*}(s)W_n^y(s)|, \tag{10}$$

where $W_n^{x*}(s)$ is the complex conjugate of the wavelet power spectrum of $x$ (Grinsted et al., 2004). The complex argument of $W_n^{xy}(s)$ gives the local phase difference between signals in $x$ and $y$ in frequency-time space. The phase relationship between the two time-series can be represented by a vector that subtends an angle representing the phase difference: On all plots of cross
spectra, arrows to the right (left) denoted signals which are in-phase and correlated (anti-correlated). Vertical arrows indicate a phase relationship of $\frac{\pi}{2}$ between the time-series, so that the evolution of one is correlated with the rate-of-change of the other. As for individual power spectra, we define a confidence interval for which cross power of a larger amplitude is deemed significant (>95% confidence interval) by comparing power exhibited by actual series with a theoretical red noise process. The cross power of two such AR1 processes is theoretically distributed such that the probability of obtaining cross power greater
than a set of red-noise processes is

$$D\left(\frac{|W_n^{xy}(s)|}{\sigma_x \sigma_y} < p\right) = \frac{Z_\nu(p)}{\nu}\sqrt{P_k^x P_k^y}, \tag{11}$$

where $\sigma$ denotes the standard deviation of the time series, Z is the confidence interval defined by $p$ ($Z = 3.999$ for 95% confidence), $\nu$ is the degrees of freedom for a real wavelet spectrum ($\nu = 2$) and $P_k^x$ is the theoretical Fourier spectrum of the AR1 process. For a given wavenumber k, this can be expressed as

$$P_k = \frac{1 - \alpha^2}{|1 - \alpha e^{2i\pi k}|^2}. \tag{12}$$



### 2.3 Hilbert Transform

We utilise a signal processing method known as a Hilbert transform to calculate the instantaneous phasor amplitude of a QBO time series. The Hilbert transform of a time series $x(t)$ can be expressed as

$$\tilde{x} = Hil[x(t)] = \frac{1}{\pi t} * x(t),$$ (13)

where ˜ denotes the transformed series, $*$ signifies a convolution and $t$ is discretised time. Conversely, the original time series can be recovered using an inverse transform expressed as

$$x(t) = Hil^{-1}[\tilde{x}(t)] = -\frac{1}{\pi t} * \tilde{x}(t).$$ (14)

A complex signal which consists of $x(t)$ and its transform is known as the analytic signal of $x$ and can be used to calculate an instantaneous phasor amplitude, $A(t)$, of the signal. $X(t)$ can be expressed as

$$X(t) = x(t) + \tilde{x}(t)i = A(t)e^{i\theta},$$ (15)

where $A(t)$ is the instantaneous amplitude of the signal and $\theta(t)$ is the instantaneous phase angle - a measure of signal progression through a cycle at time $t$.

### 2.4 Model Diagnostics

We utilise the definition of an SSW event from Butler et al. (2015). An event is recorded when the ZMZW at 60°N on the
10 hPa level transitions from westerly to easterly during NH winter months (November - March). The day on which this reversal occurs is referred to as the central date. After this date, the ZMZW must recover to westerly for a period of 10 consecutive days (which is the approximate radiative timescale of the mid-stratosphere thus allowing the vortex to recover) before another event can be recorded. If, after the central date, the ZMZW does not recover to westerly for 20 consecutive days before the end of April, the warming is classified as a final warming, when the westerly flow permanently breaks down for the season.
We analyse variability in tropical SSTs in four regions identified by Scaife et al. (2017) as key to affecting Rossby wave propagation and interactions with stratospheric winds. The regions are defined as the Tropical Atlantic ($5°S$-$5°N, 60°W$–$0°W$), Tropical East Pacific ($5°S$–$10°N, 160°$–$270°E$), Tropical West Pacific ($5°S$–$25°N, 110°$–$140°E$) and Tropical Indian Ocean ($5°S$–$10°N, 45°$–$100°E$). Additionally we calculate an El Niño Southern Oscillation (ENSO3.4) index following Trenberth and Stepaniak (2001). We use an index to track the depth of the Aleutian low pressure system as the projection of the
first Principal Component of early winter (Sep-Nov) mean sea level pressure (MSLP) anomalies averaged over the region $120°$–$240°E, 20°$–$70°N$.



Additionally we define the QBO as the monthly mean ZMZW averaged between $\pm 5°$ latitudes at various stratospheric pressure levels (15 hPa, 20 hPa, 30 hPa, 50 hPa, 70 hPa) as well as two integrated metrics as the vertical mean winds between the 15 hPa and 30 hPa levels (defined in Andrews et al. (2019)) and between 20 hPa and 50 hPa.

## 3 Results

### 3.1 Modes of Stratospheric Variability

We begin by analysing the representation of modes of stratospheric variability in the UKESM piControl simulation. As described in section 1, the winter polar stratospheric vortex exhibits substantial variability. In some years the westerly winds of the vortex are relatively strong and undisturbed while in other years the vortex is weakened by wave disturbances that in extreme cases can lead to SSWs. The average SSW rate over the full 1000 years of the UKESM simulation is 0.54 events/winter. This represents a marginal underestimation compared to ERA-Interim (0.62 events/winter between 1979 and 2019) but is within 1 standard error of the observations. The model adequately represents the seasonal distribution of SSWs compared to the reanalysis dataset, as shown in figure 1, with most SSW events occurring in February and March.

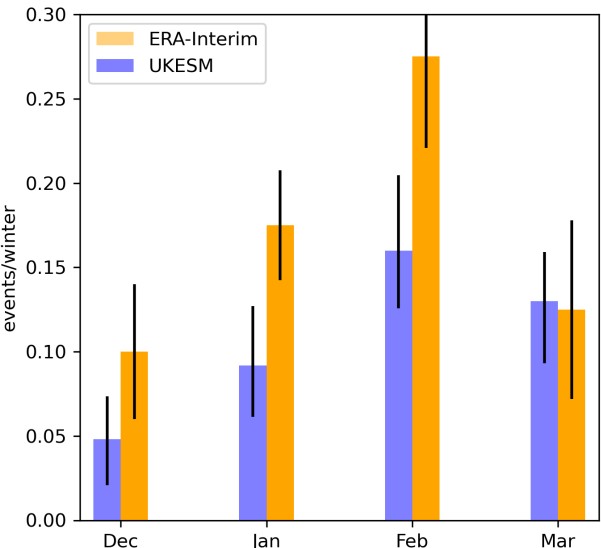

**Figure 1.** SSWs per NH winter season separated by month within the UKESM pi-control and ERA-Interim datasets. Error bars are derived using a bootstrap re-sampling method in which random selections of 50 years are chosen from the SSW data and the SSW rate recorded to build a PDF of events per season. 10000 such re-samples are carried out and the 97.5 and 2.5 percentile values are used as error bounds.



The model exhibits variability in SSW frequency comparable to observations, including hiatus intervals. Figure 2 shows a sample 40-yr interval of the polar vortex zonal wind strength from the UKESM simulation compared with a similar length from the ERA Interim reanalyses. An extended interval of mainly westerly anomalies indicating a strengthened vortex and lack of SSWs can be seen towards the end of the 40-yr interval, similar to the 1990s in ERA-Interim when only 2 SSW events were recorded in the decade. UKESM contains 8 such intervals with at least 10 consecutive years with no SSWs, the longest

of which lasts 16 years. The model time series only contains 2 intervals in which 10 consecutive years exhibit at least 1 SSW, although when the threshold interval width is shortened to 5 years 9 consecutive-event intervals and 25 hiatus intervals are found. These statistics indicate that UKESM is not only able to reproduce the mean state characteristics of SSW events but also decadal-scale variations in SSW rate, underlining its suitability for this study.

   The second major mode of stratospheric variability is the QBO at equatorial latitudes which is present at all times of the year.

Figure 3 shows the equatorial wind time-series from a sample 40-yr interval of the UKESM piControl simulation compared with the ERA-Interim dataset. The mean period of the oscillation is longer than observed, at ∼38 months compared to ∼28 months in ERA-Interim (Kawatani, 2016). As a result the vertical shear zones descend less rapidly than observed. There is also a westerly bias at low levels where the QBO-E phase does not extend sufficiently deep into the lower stratosphere, which is a common bias in many models (Bushell et al., 2020). The descending shear zones also appear more regular than observed

but there is nevertheless some evidence of decadal-scale variations e.g. in the degree of stalling at 30 hPa, although not as pronounced as in the observations.

   There is evidence of coupling between the two major modes of stratospheric variability in the model, giving rise to a Holton-

Tan relationship (Anstey et al., 2020). Figure 4 shows height-latitude cross-sections of NH winter zonal wind differences between composites defined by the QBO at various levels. The familiar pancake structure of alternating easterly / westerly differences is present at equatorial latitudes, indicative of the QBO phase but there is also a response at high latitudes. In good agreement with observations the largest high latitude response amplitude is seen when the QBO is defined at 50 hPa, with anomalously weaker polar vortex strength in QBO-E than in QBO-W years. Higher levels (15 hPa and 20 hPa) show

little significant QBO-vortex coupling. For comparison we also show in figure 4 the composite different response for QBO composites selected on the basis of the average QBO winds over a greater depth of the equatorial atmosphere (15-30 hPa and 20-50 hPa) so the impact of vertical coherence in the QBO shear zones can be assessed, following (Gray et al., 2018) and Andrews et al. (2019). Not surprisingly, the high latitude responses to the deep-QBO selections resemble the average of the individual responses but interestingly, the 15–30 hPa deep-QBO selects years that exhibit coincidence of both a weaker

polar vortex and a weaker sub-tropical tropospheric jet (see 200 hPa, 30–40N). This results in a more coherent response in the mid-latitude troposphere and at the surface, in excellent agreement with the results of Gray et al. (2018) and Andrews et al. (2019).





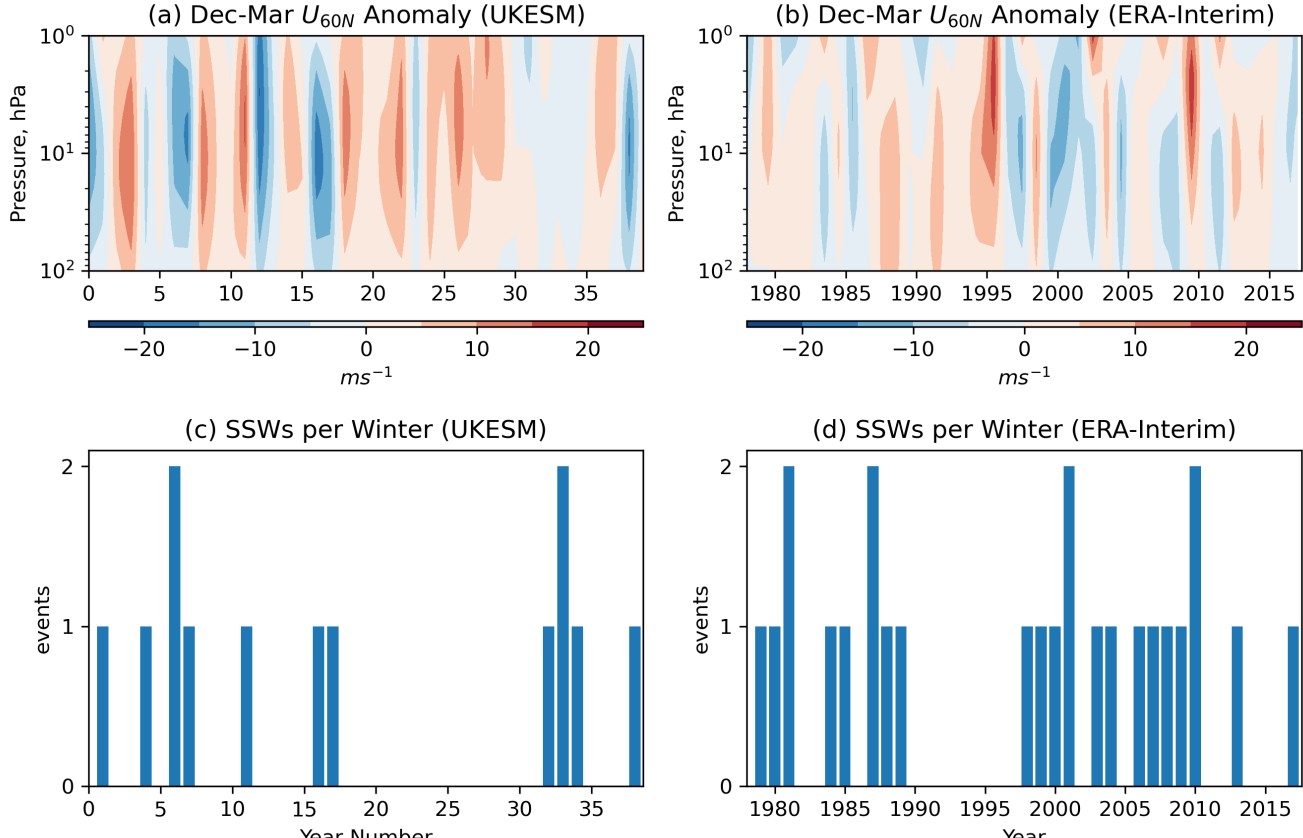

**Figure 2. (a, b)**: Dec-Mar annual mean ZMZW anomaly from the climatological mean at 60° N from a 40 year sample from the pre-industrial control simulation of UKESM **(a)** and the ERA-Interim dataset between 1979 and 2018 **(b)**. **(c, d)**: Time series of SSWs recorded per winter season in the same datasets.

The presence of the Holton-Tan relationship is also seen in the modelled frequency of SSWs (figures 5). Significantly higher

rates are observed in QBO-E winters than QBO-W. Also notable is the asymmetry in abundance of QBO-E and QBO-W winters - nearly twice as many QBO-E winters are observed compared to QBO-W under all phase definitions (figure 5, legends). This suggests an element of phase locking between the QBO and the seasonal cycle possibly associated with seasonally varying factors such as NH planetary wave forcing in winter, or advection of easterlies from the summer hemisphere (Pascoe et al., 2005; Gruzdev and Bezverkhny, 2000; Rajendran et al., 2015) resulting in QBO phase transitions that occur preferentially in

some months.



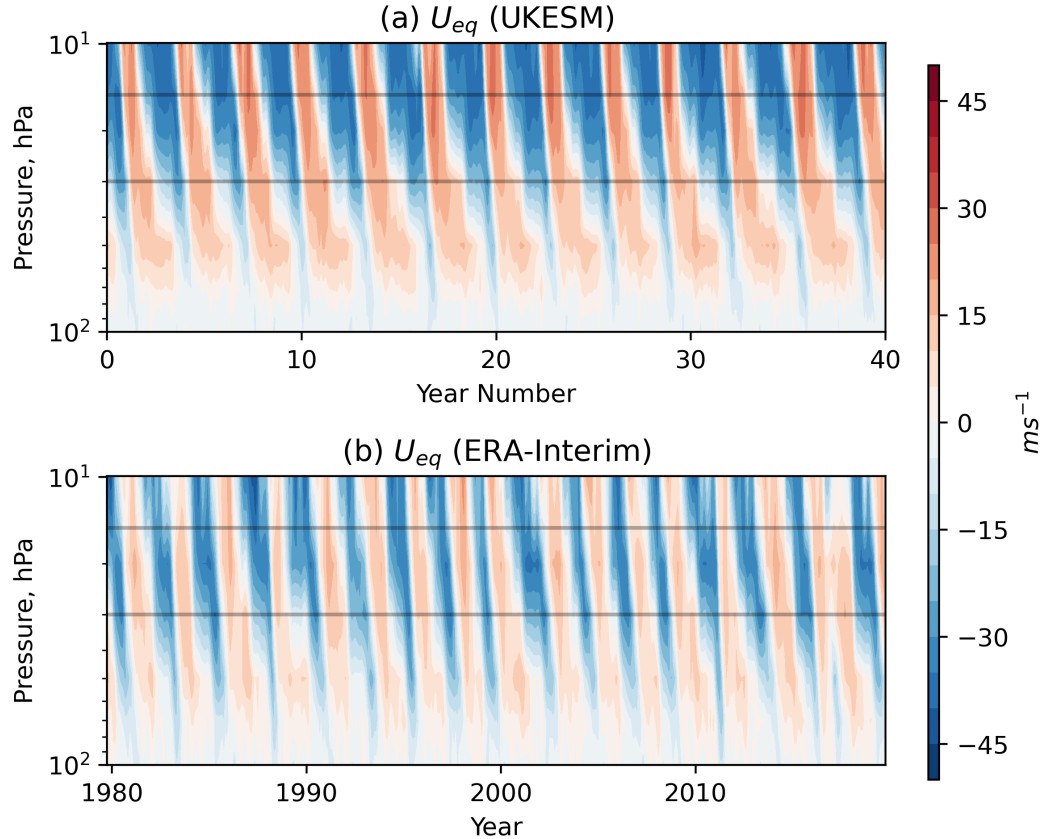

**Figure 3.** ZMZW averaged between 5° S–5° N latitude from from a 40 year sample of the pre-industrial control simulation of UKESM **(a)** and the ERA-Interim dataset between 1979 and 2018 **(b)** Horizontal lines mark the 15 hPa and 30 hPa levels between which the deep QBO metric employed by Andrews et al. (2019) is defined.

## 3.2 Long-term Variability of the Polar Vortex

A more comprehensive assessment of the long-term variability of SSWs can be made using a wavelet power spectrum approach. We count the number of SSWs in each winter season (Dec-Mar) and calculate the corresponding wavelet power spectrum 275 (figure 6). As expected, there is a QBO-like signal of period around 2-4 years, confirming the presence of a Holton-Tan relationship between the QBO and the polar vortex in the model. The signal is intermittent throughout the simulation and the global spectrum (the time average of the wavelet spectrum, shown on the right of the figure) shows that the signal is on the boundary of assigned statistical significance. Other signals at periods near 20-30 years are similarly intermittent and manifest as a peak in the time-averaged spectrum that is also near the assigned significance boundary. The most persistent feature of the 280 series appears at periods between ∼60-90 years in the interval between 400-800 yrs. This feature shows statistical significance





**Figure 4.** Dec-Mar ZMZW composite differences between QBO East and QBO West phases evaluated in Sep-Oct at individual levels as well as using the deep QBO metric. The phase of the QBO is defined as in figure 1 - the equatorial Sep-Nov ZMZW of greater magnitude than $5\,\mathrm{m\,s^{-1}}$. Coloured shading indicates differences significant above the 95% confidence level under a 2 tail student's t-test.

(based on comparisons between power in the spectrum and that of an AR1 process with the same autocorrelation structure as the series being analysed) for around 350 years of the 1000-yr simulation but does not cross the significance threshold for the time-averaged spectra. There is a possible limitation of this wavelet methodology due to the discrete nature of the time series being analysed (time points take values 0/1/2). The Morlet wavelet is a continuous function and, as a result, convolution with a

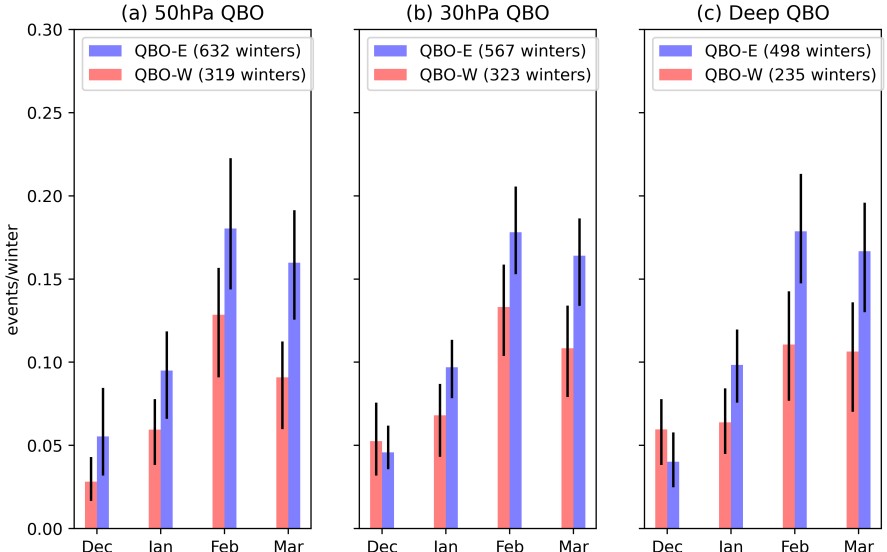

**Figure 5.** SSWs per winter season for years exhibiting QBO-E and QBO-W conditions in early winter (Sep-Nov) defined on different pressure levels (a,b) as well as using the deep metric (c), the vertical mean between 15 and 30 hPa defined in Andrews et al. (2019). The QBO phase is defined as any Sep-Nov equatorial ($5°$ S–$5°$ N average) ZMZW that exceeds a magnitude of $5\,\mathrm{m\,s^{-1}}$. Error bars on all plots are derived using the same bootstrapping method outlined in figure 1.

highly discretised series may alias features on the resulting wavelet spectra. This limitation must be considered when drawing conclusions from the wavelet spectra and is discussed further below.

     The focus of this study is on the longer-term time variations to understand the source of variability characterised by hiatus intervals (no SSWs over an extended period) and consecutive-event intervals (at least one SSW every year for an extended

period). We therefore apply low-pass filtering to the time series of SSWs per season using a 5-year rolling window and examine the spectral characteristics of this smoothed series (which we refer to henceforth as $SSW_{5yr}$). This averaging is similar to the standard practice of smoothing daily data to remove the noise associated with daily weather variations, thus isolating longer seasonal timescales. It also decreases the impact of the time series discretisation by reducing the chance of introducing spurious spectral features on the wavelet power spectrum which may be otherwise encountered when analysing the un-smoothed time

series.

     The wavelet power spectrum of $SSW_{5yr}$ (figure 7) shares many of the characteristics of the spectra of the un-smoothed series (figure 6), but the longer period signals are now more clearly evident, as expected. The $SSW_{5yr}$ wavelet spectrum shows the

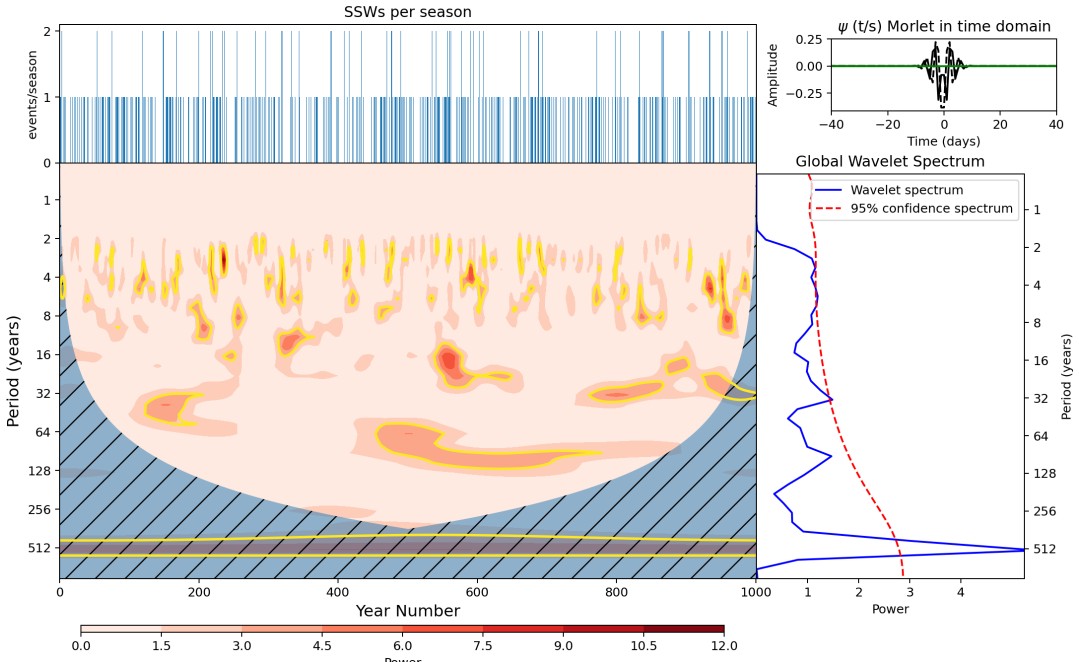

**Figure 6. Top left**: SSW events per Dec-Mar season in UKESM **Bottom left**: Wavelet power spectrum of time series in top left. Hatching represents area outside the cone of influence in which edge effects are significant and power should not be considered. Yellow contours represent the 95% confidence level assuming mean background AR1 red noise. **Top Right**: Morlet wavelet used for the wavelet transform in the time domain. **Bottom right:** Global power spectrum, the wavelet power averaged over the whole simulation, and global 95% confidence spectrum.

two broad regions of statistically significant maxima corresponding to signal periods of ∼20-30 years and ∼60-90 years, but with increased significance both locally and in the time-average. For example, the feature around 90 year period appears significant for 450 years in $SSW_{5yr}$ compared to 350 years before smoothing. One possible explanation for this increase lies in our definition of the significance level on power which is dependent on the lag-1 autocorrelation of the time-series. Introducing a 5 year averaging window will increase the autocorrelation, possibly leading to a less strict significance level. However, this is unlikely because the significance level is constructed using a red noise process with the same autocorrelation as the series. This means that for $SSW_{5yr}$, the threshold for 95% confidence level increases with increasing period more steeply than in the un-smoothed case and yet the power exhibited at those long periods in $SSW_{5yr}$ nevertheless achieves higher statistical significance. This indicates that the smoothing has enhanced the visibility of a real signal in the $SSW_{5yr}$ time series that was less visible in the un-smoothed time-series.



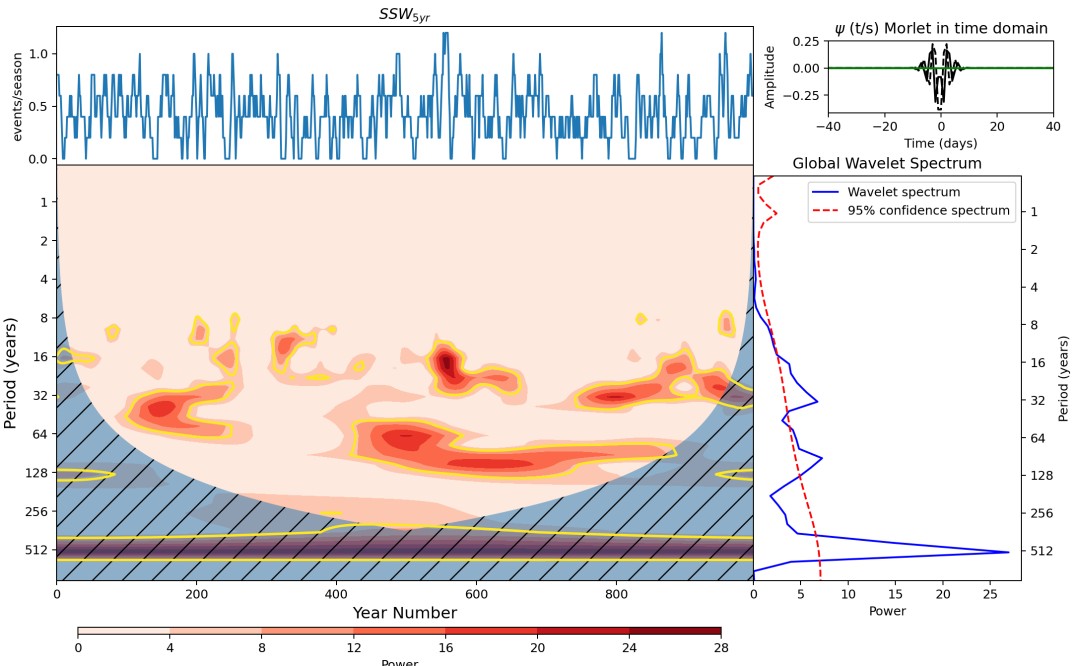

**Figure 7. Top left**: SSW events per Dec-Mar season in UKESM smoothed using a 5 year running mean. **Bottom left**: Wavelet power spectrum of time series in top left. Hatching represents area outside the cone of influence in which edge effects are significant and power should not be considered. Yellow contours represent 95% confidence level assuming mean background AR1 red noise. **Top Right**: Morlet wavelet used for the wavelet transform in the time domain. **Bottom right:** Global power spectrum, the wavelet power averaged over the whole simulation, and global 95% confidence spectrum.

### 3.3 Surface Forcing of Polar Vortex Variability

In the absence of external forcing mechanisms such as greenhouse gas or anthropogenic aerosol forcing, the presence of long-term variability such as the 60-90 year periodicity seen in $SSW_{5yr}$ (figure 7) suggests a source of long-term internal variability from within the climate system.

  The most obvious potential driver of such long timescale variability is the ocean due to its high degree of thermal inertia. Previous work has identified coupling between tropical SSTs and the polar vortex, such as the relationship to ENSO conditions 315 (see section 1). Figure 8A shows the wavelet power spectrum for the 5 year smoothed Sep-Nov ENSO 3.4 index as well as the cross power spectrum with $SSW_{5yr}$. We use the early NH winter ENSO index to capture the lagged response of the vortex to this mode of variability. The ENSO index is slowly varying so will likely remain in the same state between early and mid-winter. We also smooth the ENSO index for the purposes of calculating the cross spectrum with $SSW_{5yr}$. (The spectrum of the un-smoothed ENSO 3.4 index is provided in supp figure A1 and shows significant power in the expected period range of 320 4-7 years (Santoso et al., 2017)). The smoothed ENSO 3.4 index shows intermittent power at periods around 16 years which





appears significant in the global spectrum. It also exhibits a small signal coincident with the 90 year variability in $SSW_{5yr}$, however this feature only persists for around 100 years of the simulation. Cross spectra between the two series (figure 8B) reveals that the coincidence in signals at the 90 year period, while significant under our test, is marginally prominent but only covers a small proportion of significant signals in $SSW_{5yr}$. This suggests there may be some contribution from ENSO to

the observed SSW variability but it is only marginally significant and on its own it cannot explain the signal in $SSW_{5yr}$ that persists for 450 years.

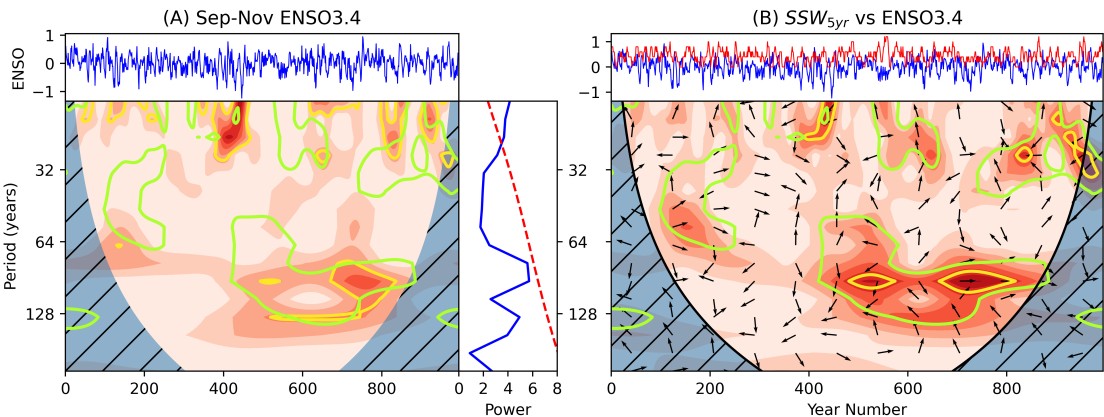

**Figure 8. (A, top)**: ENSO3.4 time series, **A, bottom left**: Wavelet power spectrum (shaded contours represent wavelet power and yellow contours the 95% significance level compared to an AR1 process), **A, bottom right**: global wavelet power spectrum (blue) and 95% confidence level (dashed red). **B**: Cross spectra between $SSW_{5yr}$ and the ENSO3.4 index. **B, top**: ENSO3.4 and $SSW_{5yr}$ time series. **B, bottom**: Cross power spectrum. Shading indicates cross power, yellow contours the 95% confidence interval and arrows the relative phase angle between signals in the time series (to the right: in phase, vertically upwards: $\frac{\pi}{2}$ out of phase with SSWs leading, to the left: $\pi$ out of phase, vertically downwards: $\frac{\pi}{2}$ out of phase ENSO3.4 leading). Green contours on both spectra represent the 95% confidence intervals for the wavelet power spectrum of $SSW_{5yr}$.

In the interest of completeness, we also explore the long-term variability of other tropical ocean regions and their potential teleconnections with the polar vortex. Four additional tropical regions were selected based on those identified by Scaife et al.

(2017) and outlined in section 2. While all four regions show some elements of multi-decadal variability (supp figure A2), particularly the Tropical Atlantic with a peak period of approximately 140 years for 700 years of the simulation, none of the spectra show variability that coincides well with that of $SSW_{5yr}$. There is some overlap of the Atlantic and Tropical East Pacific spectra with the regions of significant periodicity at around 60-90 years in the $SSW_{5yr}$ spectrum but, like the ENSO3.4 index, the overlaps and cross power between the series (supp figures A2 and A3) are minimal and cannot reasonably explain

the vortex signal, especially the signal of period approximately 90 years that persists in $SSW_{5yr}$ for around 450 years (supp figures A2 and A3, green contours).





The strength of the Aleutian Low (AL) has also been used as an indicator of tropospheric wave forcing and its influence on the polar vortex (Woo et al., 2015). A similar wavelet and cross-spectrum analysis was therefore performed using an index based on the strength of the modelled NH winter (Dec-Mar) AL (see section 2 for details). The wavelet power spectrum for the

5-year smoothed AL index (figure 9A) exhibits elements of periodic signals with maximum power corresponding to a period of around 55 years (between 40-60 years) but with fairly minimal overlap with the regions enclosed by the 95% confidence level in the corresponding SSW wavelet analysis (green contours). The cross spectrum analysis between the AL and $SSW_5yr$ (figure 9B) highlights this relatively small region of overlap in the interval between years 400-500. However, the indicated phase relationship in that region of overlap is difficult to interpret. The proposed physical mechanism of coupling between

the AL and the vortex (Woo et al., 2015) involves an association between a deeper AL (i.e. lower pressure) with increased frequency of SSWs. This negative correlation would give rise to arrows pointing to the left if the relationship was present. In contrast, the upward arrows figure 9B indicate a $\frac{\pi}{2}$ phase difference between the indices on these 60 year timescales, suggesting that peaks in $SSW_{5yr}$ variations are associated with maximum rates of change of the AL index at the same periods. Given the relatively short time interval of overlap between the AL and $SSW_{5yr}$ signals at the 60-yr period, the absence of any

significant signal around the 90-yr period, together with the inconsistent phase relationships noted above, we conclude that the long-term variability in $SSW_{5yr}$ is unlikely to be associated with AL forcing in this simulation. Indeed, examination of the cross-spectrum between the un-smoothed AL and SSWs per season indices (figure A4) shows no indication of a coherent relationship between the two indices at any timescale, and the Pearson correlation between the two indices is only 0.21.

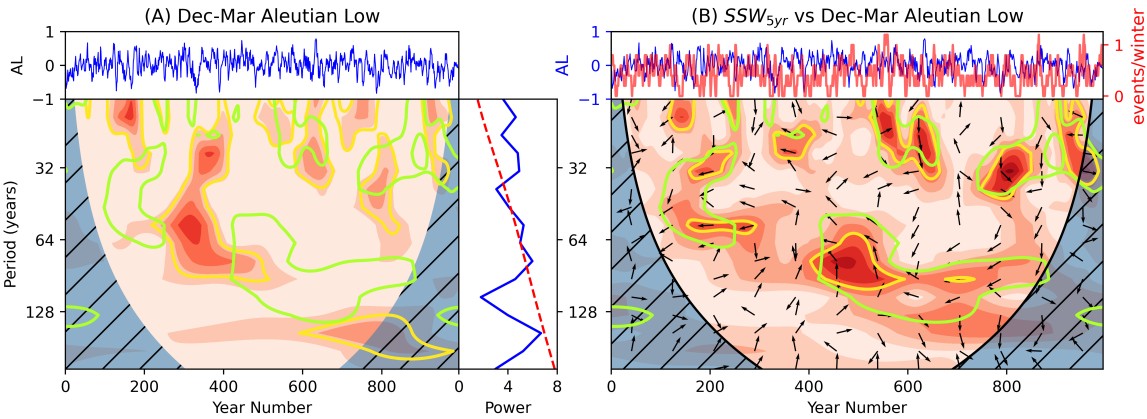

**Figure 9.** As figure 8 for the Dec-Mar Aleutian Low index smoothed with a 5 year window. **A**: AL time series and associated wavelet power spectrum. **B**: Cross power spectrum between AL and $SSW_{5yr}$.





### 3.4 Vortex-QBO Interactions

Despite some coincident signals between tropical SSTs, AL and $SSW_{5yr}$, long-term variability in these surface indices are unable to fully account for the multidecadal signals in SSW frequency. An additional potential source of internally generated long-term variability may reside within the stratosphere. Studies have noted relatively long-term variations in the strength of the Holton-Tan relationship (Lu et al., 2008, 2014; Osprey et al., 2010) although the cause of these variations is not well understood. In order to investigate this figure 10 shows the wavelet power spectrum of early winter (Sep-Nov) QBO winds evaluated at selected levels. Since the QBO evolves relatively slowly, employing Sep-Nov averaged winds provides a reasonable representation of the QBO and also allows us to evaluate the in-season lagged relationship between the QBO and subsequent occurrence of an SSW. There is a clear signal between 2 and 4 years for the majority of the simulation, as expected, but no prominent power at longer periods, confirming that there is no significant long-term variability in the periodicity of the QBO winds that could explain the long-term variations in $SSW_{5yr}$ via the Holton-Tan relationship.

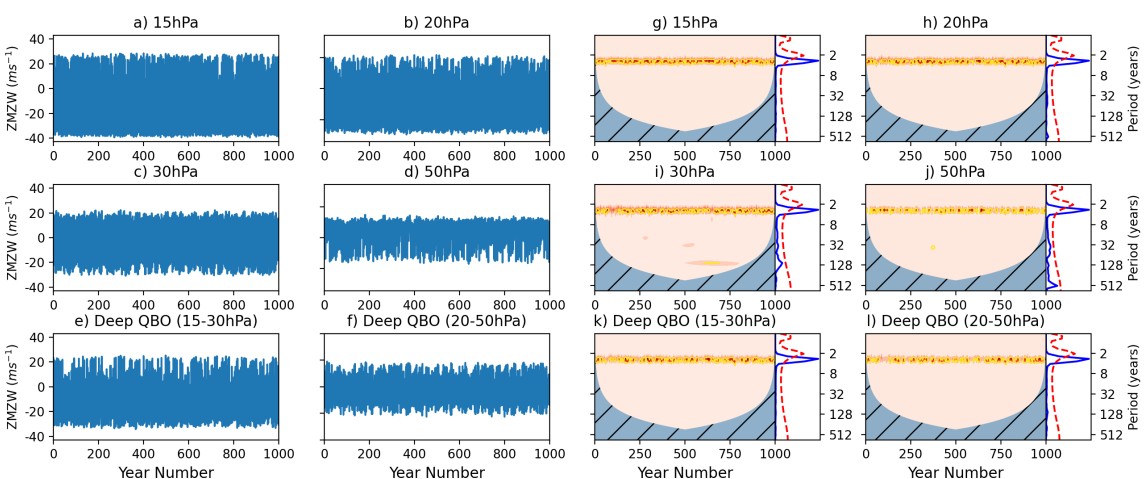

**Figure 10. (a-f)**: Sep-Nov mean ZMZW averaged between 5°S–5°N latitude on different pressure levels (a-e) and the deep metric averaged between 15–30 hPa (f). **(g-l)**: Wavelet power spectra for each time series shown in (a-c). Shading represents wavelet power and yellow contours indicate regions of significant power (>95% confidence interval) compared to a background AR1 process.

While the wavelet analysis technique is able to isolate and reveal frequency modulations very well it is less suited to examine amplitude modulations which are clearly evident by eye in some of the QBO index time-series. For example, both the 20 hPa and deep (15-30 hPa) QBO time series show multi-decadal variations in the magnitude of the westerly phase while the easterly phase amplitudes are relatively uniform in time. Similarly, the 50 hPa and 30 hPa time series show amplitude modulation predominantly in the easterly phase. This amplitude modulation can be highlighted by taking the Hilbert Transform of each QBO





time-series (figure 11a-f). Wavelet analysis of the transformed QBO time series now shows significant power on multidecadal timescales (figure 11g-l). In particular, the 20 hPa and deep QBO time-series exhibit signals coincident in time and around similar periods (60-90 years) to those observed in $SSW_5yr$. On the other hand, the QBO indices based on equatorial winds at
50 hPa or 30 hPa show minimal power at these periods, despite showing a strong intraseasonal HT relationship (figure 4). Given that the 15-30 hPa deep QBO index exhibits both multidecadal timescale variability and a strong intraseasonal HT coupling, we continue further analysis of the SSW-QBO interactions using the 15-30 hPa index.

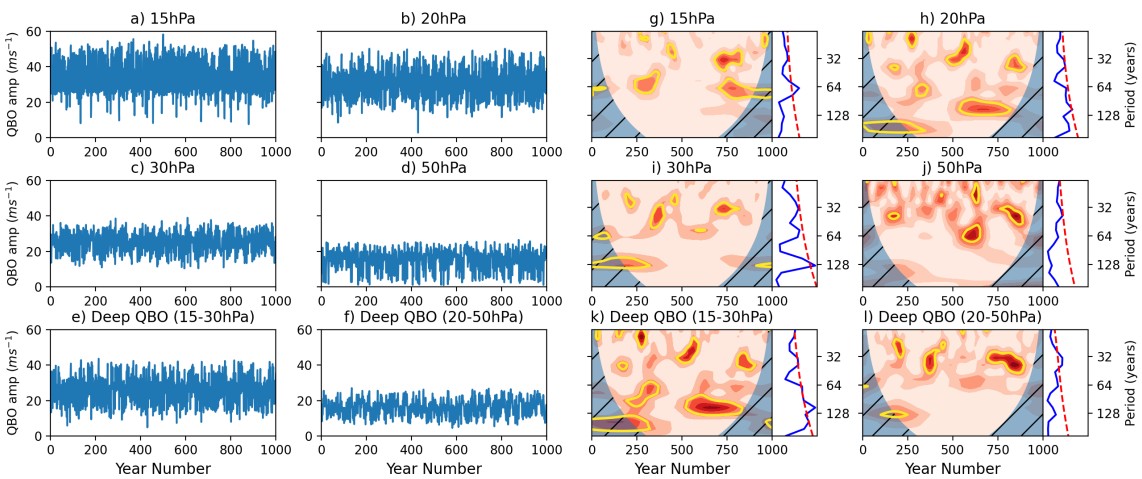

**Figure 11. (a-e)**: Hilbert amplitude of Sep-Nov mean ZMZW averaged between 5° S–5° N latitude on different pressure levels (a-e) and the deep metric averaged between 15–30 hPa (f). **(g-l)**: Wavelet power spectra for each time series shown in (a-c). Shading represents wavelet power and yellow contours indicate regions of significant power (>95% confidence interval) compared to a background AR1 process.

     Wavelet analysis of the 5-year smoothed deep (15-30 hPa) QBO amplitude modulation index (figure 12A) enhances the
clarity of the long-term periodicity, showing statistically significant power at around 90 years in the interval between 500-800 years. The cross power between $SSW_5yr$ and this QBO amplitude modulation index (figure 12B) coincides extremely well with the signals observed in $SSW_5yr$ at around 90 years. There are also coincident features at other timescales, although the feature between years 450-550 at periods of 60 years is less well captured. The phase-relationship arrows in the main region of long-term variability (periods around 90 years in the interval 450-800 years) point broadly to the left ($\pi$ phase shift), indicating
that the signals are approximately anti-phased (the slight downward pointing of the arrows suggests a small deviation from this lag-zero relationship and is discussed below). The anti-phase relationship is consistent with the HT relationship in which a westerly (positive) QBO anomaly corresponds to a reduction in the frequency of SSWs.



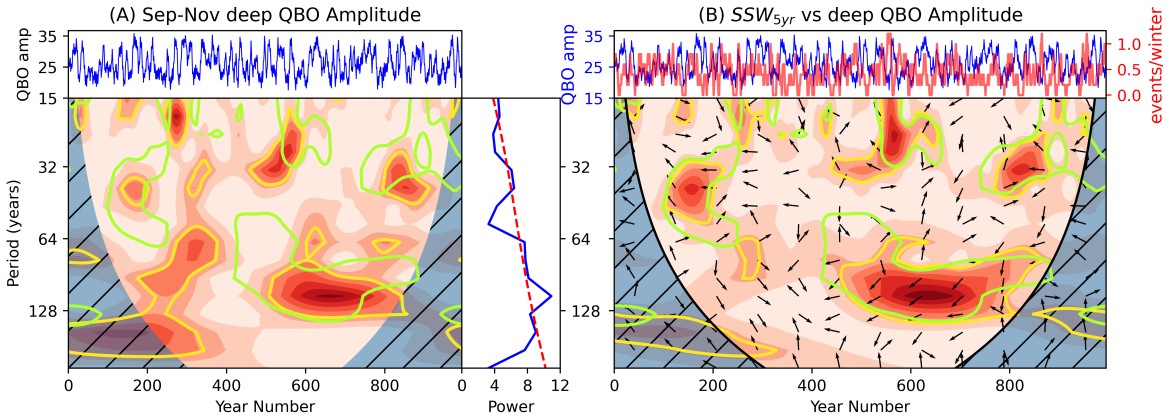

**Figure 12.** As figure 8 for the Sep-Nov deep QBO (15–30 hPa) amplitude index smoothed with a 5 year window. **A**: QBO amplitude time series and associated wavelet power spectrum,**B**: Cross power spectrum between deep QBO amplitude and $SSW_5yr$.

In our earlier discussion we linked long-term variability in SSW frequency to the existence of extended hiatus periods, during which the vortex is relatively undisturbed with no SSW events (figure 2). The cross-spectrum analysis with deep QBO amplitude modulation suggests a possible physical interpretation involving the Holton-Tan relationship acting on longer timescales, in which a series of consecutive years that exhibit a large amplitude, deep westerly QBO in early winter leads to a series of winters with reduced SSW frequency i.e. a hiatus period. Correspondingly a series of large-amplitude deep easterly QBO years would lead to a series of consecutive-event years.

To further clarify the role of the QBO, we note that an examination of figure 10 shows that the majority of the long-term amplitude variability in the 15-30 hPa deep QBO index lies in the amplitude of the westerly phase (the easterly phase amplitude is relatively constant with time). Also, as noted earlier, the simulation exhibits more hiatus intervals than consecutive-event intervals, which suggests that the observed long-term variability may arise primarily from the westerly QBO phase. To explore this hypothesis, we isolate the SSW hiatus intervals by modifying the $SSW_5yr$ time-series in the following way. All SSW rates above 0.54 events per season (the climatological mean) are re-set to 0.54 thereby removing variability in 5 year intervals that exhibit anomalously high SSW rates. Figure 13 shows the cross power spectrum between this modified $SSW_5yr$ time-series and the time-series of deep QBO amplitude. It retains significant cross power within the portion of significant $SSW_5yr$ power (figure 13 green contours) when compared with figure 12 in which the full time-series is used and also shows a phase relationship significantly closer to anti-phased. This confirms that the deep QBO - SSW relationship on these long timescales arises primarily from the SSW hiatus periods.

An obvious question is whether this sensitivity to deep QBO westerlies that we find in the model is also present in the real atmosphere. Examination of the ERA-Interim dataset shows limited support. Some winters in the 1990s are characterised by



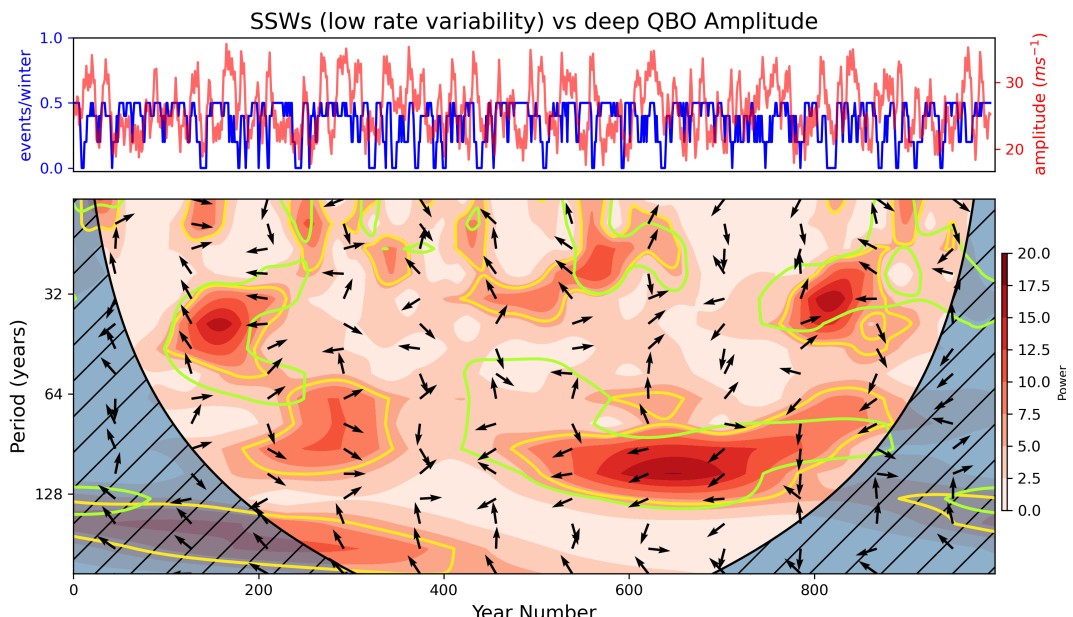

**Figure 13. Top**: $SSW_{5yr}$ time series with variability in high SSW rate intervals removed by setting all rates above the climatological mean (0.54 events per season) to the mean (blue) and Sep-Nov deep QBO Hilbert Amplitude index smoothed with a 5 year window (red). **Bottom**: Cross wavelet power spectrum between the two time series.

anomalously westerly Sep-Nov equatorial winds which are vertically coherent between the 15 and 30 hPa levels. However this effect is intermittent and does not span the whole interval of the 1990s during which SSWs were markedly absent in the observational record. On the other hand, a mini hiatus in the mid 2010s is associated with 3 years of deep westerly anomaly in the QBO. Overall, it is clear that the signal, if present, is likely obscured by other factors including greenhouse gas increases and volcanic eruptions and the observational dataset is too short to provide useful validation for these long timescale variations.

## 4  Summary and Discussion

In this study, we have examined variability in the appearance of hiatus and intervals of consecutive SSWs in a 1000-yr pre-industrial control simulation of the UK Earth System Model. While there is much observational evidence for an impact of SSWs on the underlying tropospheric weather and climate the observational record is too short to draw robust conclusions about their multi-decadal variability or the associated forcing mechanisms. Analysis of long climate model simulations is currently the only available tool for understanding this variability.

We found realistic decadal and multi-decadal variability in the model, with hiatus intervals of 10 years or more in which no SSWs occurred, similar to the observed SSW record in the 1990s (Pawson and Naujokat, 1999; Shindell et al., 1999) and also intervals of consecutive-event periods in which at least one SSW occurred every year, as observed in the early 2000s (Manney





et al., 2005). A 5-yr smoothed representation of SSW frequency ($SSW_{5yr}$) was found to vary periodically for approximately 450 years of the 1000-yr simulation with maxima in wavelet power corresponding to periodicity of around 60-90 years.

A possible tropical SST source of this long-term variability was investigated. Wavelet and cross-spectrum analyses were performed using a variety of different tropical SST indices, including the ENSO 3.4 index, and also an index of the strength of the Aleutian Low (AL) which is linked to large-scale planetary wave forcing of the winter stratosphere. While all of these indices displayed long-term variability, some of which overlapped with the periodicity and time intervals seen in the $SSW_{5yr}$ spectrum, none of them could fully account for the extended 450 year interval of significant power at 60-90 years seen in the

$SSW_{5yr}$ spectrum.

  A second possible source of long-term variability involving variations in the QBO was also investigated. A range of QBO indices were considered, including the standard approach of using equatorial winds at a specified pressure level e.g. 50 hPa and also a 'deep QBO' index which takes the average QBO wind over 15-30 hPa, designed to capture the degree of vertical coherence in the QBO winds (following Gray et al. (2018) and Andrews et al. (2019)). A straightforward wavelet analysis

of these QBO indices reveals no power at periodicity longer than 2-4 years. However, while there is evidently no long-term variability in the frequency of the QBO, visual examination of the QBO time series clearly shows the presence of long-term variability in the QBO amplitudes. A measure of this amplitude modulation was extracted by taking the Hilbert Transform of the QBO index. Wavelet analysis of the amplitude variations from the Hilbert Transform of the QBO indices showed long-term periodicity matching that seen in the $SSW_{5yr}$ wavelet analysis. The deep QBO index showed greatest overlap with the

$SSW_{5yr}$ power spectrum, particularly at periodicities of around 90 years. This overlap accounted for nearly all 450 years of SSW variability present on the 90 year timescale.

  Our analysis has therefore revealed an unexpected relationship between the strength and vertical coherence of the QBO and long-term variability in the frequency of SSWs. The relationship was found to be particularly sensitive to the QBO westerly phase. Extended periods of deep westerly QBO phases were associated with hiatus periods with few or no SSWs, consistent

with the Holton Tan relationship.

  Combining the results of all these analyses, our overall conclusion is that multi-decadal variability in SSW frequency in the model is primarily accounted for by long term variability in QBO-SSW coupling, particularly at periodicities of around 90 years and, to a lesser extent, by variability in the depth of the Aleutian Low at periodicities around 60 years, although coherence with the AL signals is far less persistent than with the QBO. Given the observed impact of SSWs on the underlying

tropospheric weather and climate, improved understanding of the source and mechanisms of long-term variability in QBO-SSW interactions is likely to help improve future seasonal weather forecasts and decadal-scale climate predictions. The precise nature of QBO-SSW interaction mechanisms are still not fully understood (Anstey and Shepherd, 2014). While the importance of wave-mean flow interactions is widely recognised, further studies are required to explore the relevance and usefulness of the deep QBO index highlighted in this study, that identifies a vertically coherent QBO phase. It appears to be especially relevant

to long-term QBO-SSW interactions during the QBO-W phase.

  Further exploration of the source of the long-term variability in amplitude of the QBO-W phase is also required. While a direct influence of tropical SSTs on long-term variability in SSW frequency has not been supported by this study there may





nevertheless be an important teleconnection via the QBO, in which the SSTs influence the QBO which subsequently influences the SSW frequency via the Holton Tan relationship. Initial investigation through cross spectrum analysis of the deep QBO
index with ENSO and selected SST indices shows some contribution from each of the regions (supp figures A5 and A6), not surprisingly because of the tropical source of equatorial waves that are known to drive the QBO. A closer examination of the precise nature of forcing of the QBO-W phase in the model, in terms of Kelvin and gravity wave forcing would be helpful (but outside the scope of this study).

Time series analysis such as those presented here can highlight associations between modes of variability but are less able
to determine causality. Where possible we have selected indices that confirm well-known in-season causality, such as an early winter QBO index compared with a mid-winter SSW index, but determining causality on longer timescales is much more difficult. Well designed climate model experiments can help, for example to eliminate SST variability by using an atmosphere-only model with climatological SST forcing which could help to identify whether the QBO-W amplitude variations are externally forced by the SSTs or whether they are internally-driven (e.g. through nonlinear interactions within the stratosphere). Similarly,
there are other potentially important teleconnections that have not been examined in this study, for example the possible role of North Atlantic and/or North Pacific blocking, and other potential sources of long-term variability such as the Pacific North America (PNA) pattern, the Pacific Decadal Oscillation (PDO) and the Atlantic Multidecadal Variability (AMV) index. The climate system is extremely complex, with many different interactions between these modes of variability. The climate system is also clearly non-stationary, as evident in our simulation where the QBO-SSW interaction shows power at periodicities of
60-90 years for 450 years but is absent in the early half of the simulation. While this complexity means that it is extremely challenging to disentangle the influences or to attribute causality, improved understanding of individual links in this complex system, such as the relationship between the QBO and SSWs, will nevertheless contribute to improved understanding of the whole complex system.



*Data availability.* ERA-Interim reanalysis data are available from the Copernicus Climate Change Service Climate Data Store
(CDS, https://climate.copernicus.eu/climate-reanalysis, C3S, 2017). Data from the UKESM simulation used in this study are
available from the Earth System Grid Federation of the Centre for Environmental Data Analysis
(ESGF-CEDA;https://esgf-index1.ceda.ac.uk/projects/cmip6-ceda/,WRP,2019, last access: 6 Oct 2020).

*Author contributions.* OBDM conducted the analyses, and LJG and SO directed the research. All authors were fully involved
in the revisions and the preparation of the paper.

*Competing interests.* The authors declare that they have no conflict of interest.

*Financial support.* LJG and SO acknowledge funding from the UK Natural Environment Research Council (NERC) through
the National Centre for Atmospheric Science (NCAS) ACSIS project (NE/N018001/1) and the NERC Belmont-Forum grant
GOTHAM (NE/P006779/1). OBDM gratefully acknowledges support from the Oxford DTP in Environmental Research
(NE/L002612/1)

*Acknowledgements.* The authors would like to thank their respective funding bodies as well as Tim Woollings, Antje
Weisheimer and Chris O'Reilly for useful discussions. We also give thanks to the UKESM1 team who have worked to
develop and run the model used in this study as well as make the data available. In particular, Colin Jones, Jeremy Walton,
Including results for Alistair Sellar and Till Kuhlbrodt.



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




## Appendix A

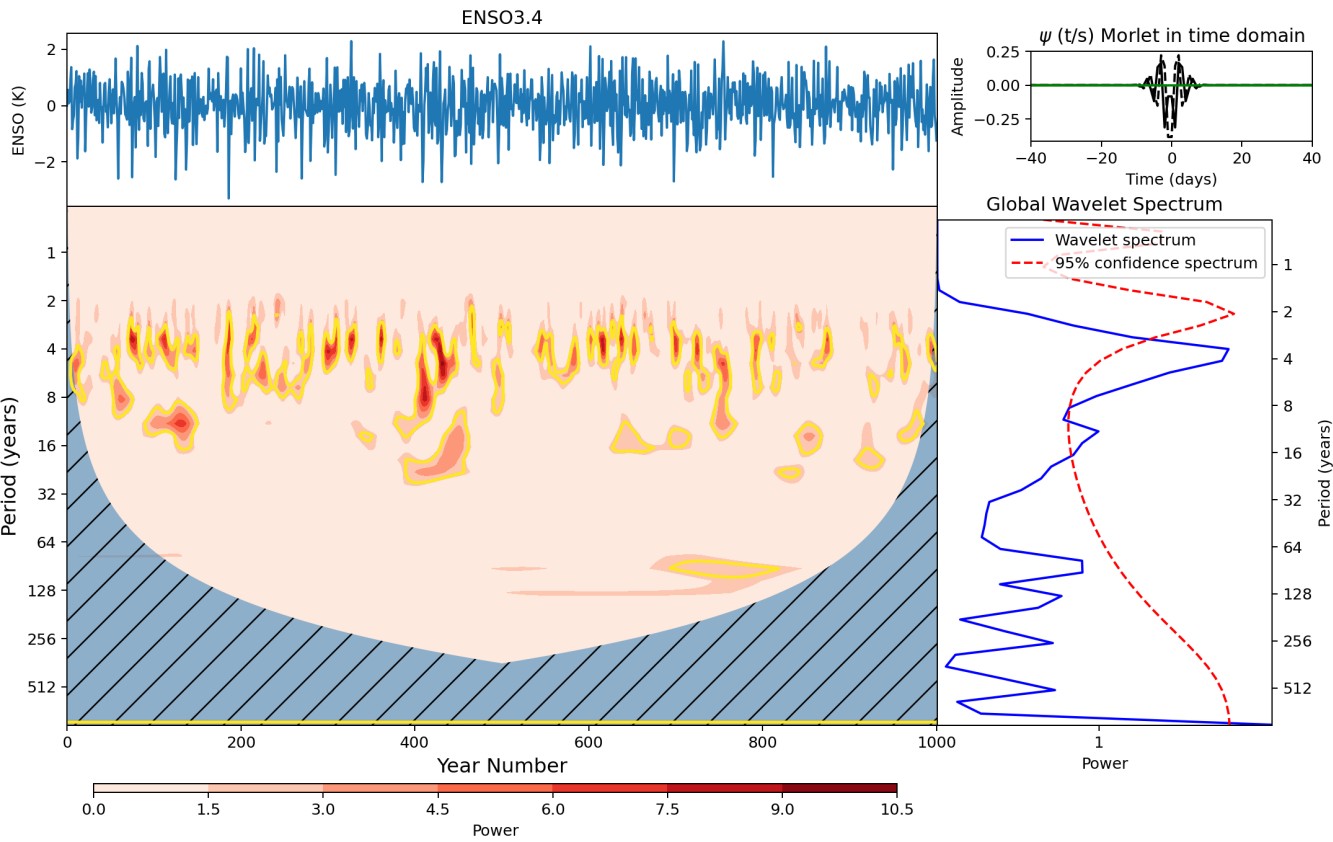

**Figure A1. Top left**: Sep-Nov ENSO3.4 index from the UKESM pi-control simulation. **Bottom left**: Wavelet power spectrum of time series in top left. Hatching represents area outside the cone of influence in which edge effects are significant and power should not be considered. Yellow contours represent the 95% confidence level assuming mean background AR1 red noise. **Top Right**: Morlet wavelet used for the wavelet transform in the time domain. **Bottom right:** Global power spectrum, the wavelet power averaged over the whole simulation, and global 95% confidence spectrum.

680



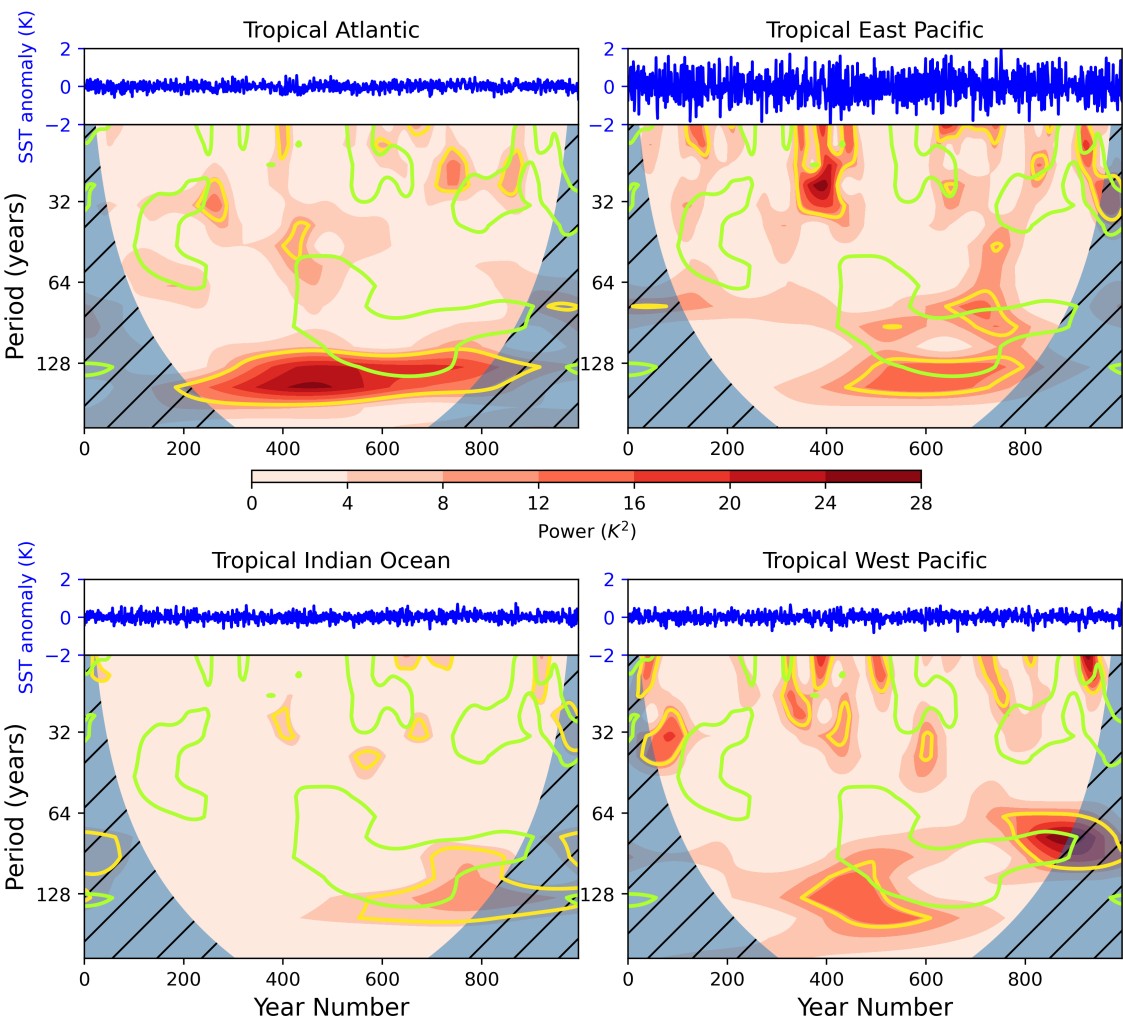

**Figure A2.** Sep-Nov SST anomaly time series and associated wavelet power spectrum for Tropical Atlantic (5° S–5° N, 60° W–0° W), Tropical East Pacific (5° S–1° N, 160° E–270° E), Tropical West Pacific (5° S–25° N, 110° E–140° E) and Tropical Indian Ocean (5° S–10° N, 45° E–100° E). Shading indicates wavelet power, yellow contours show the 95% confidence level when the power is compared to and AR1 red-noise process and green contours indicate the 95% confidence level for the power spectrum of $SSW_5yr$.

685

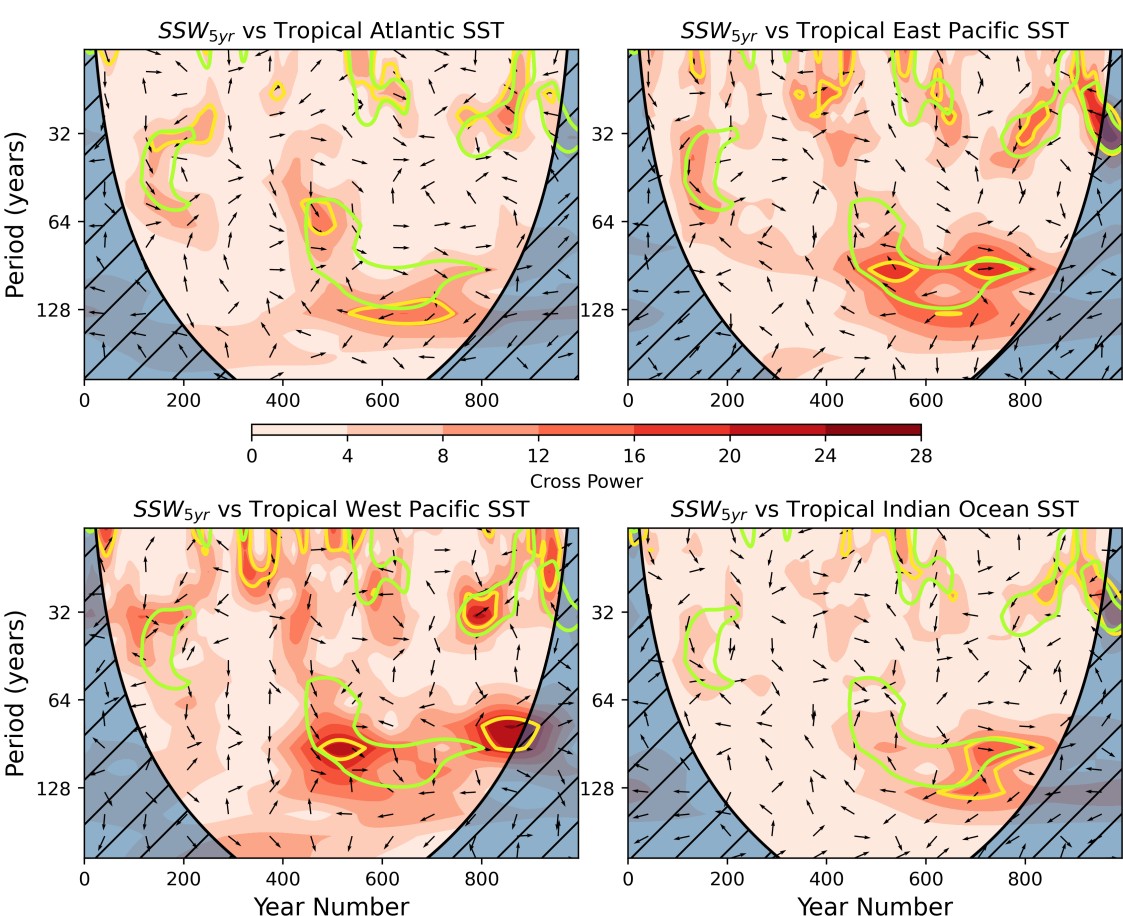

**Figure A3.** Cross power spectra between Sep-Nov Tropical SST anomaly time series and $SSW_5yr$. SST regions are defined as: Tropical Atlantic (5° S-5° N, 60° W-0° W), Tropical East Pacific (5° S–1° N, 160° E–270° E), Tropical West Pacific (5° S–25° N, 110° E–140° E) and Tropical Indian Ocean (5° S–10° N, 45° E–100° E).

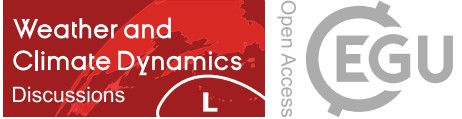



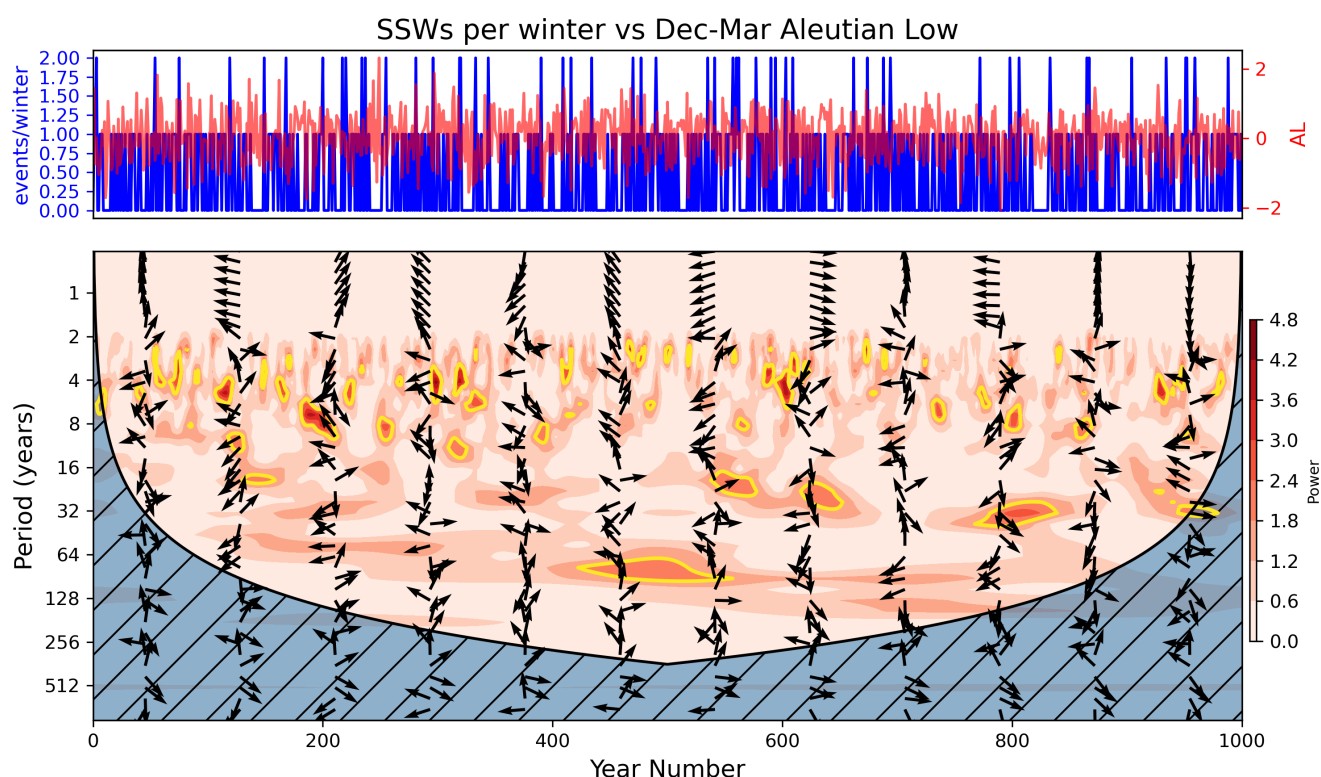

**Figure A4. Top left**: Dec-Mar Aleutian Low index from the UKESM pi-control simulation. **Bottom left**: Wavelet power spectrum of time series in top left. Hatching represents area outside the cone of influence in which edge effects are significant and power should not be considered. Yellow contours represent 95% confidence level assuming mean background AR1 red noise. **Top Right**: Morlet wavelet used for the wavelet transform in the time domain. **Bottom right:** Global power spectrum, the wavelet power averaged over the whole simulation, and global 95% confidence spectrum.



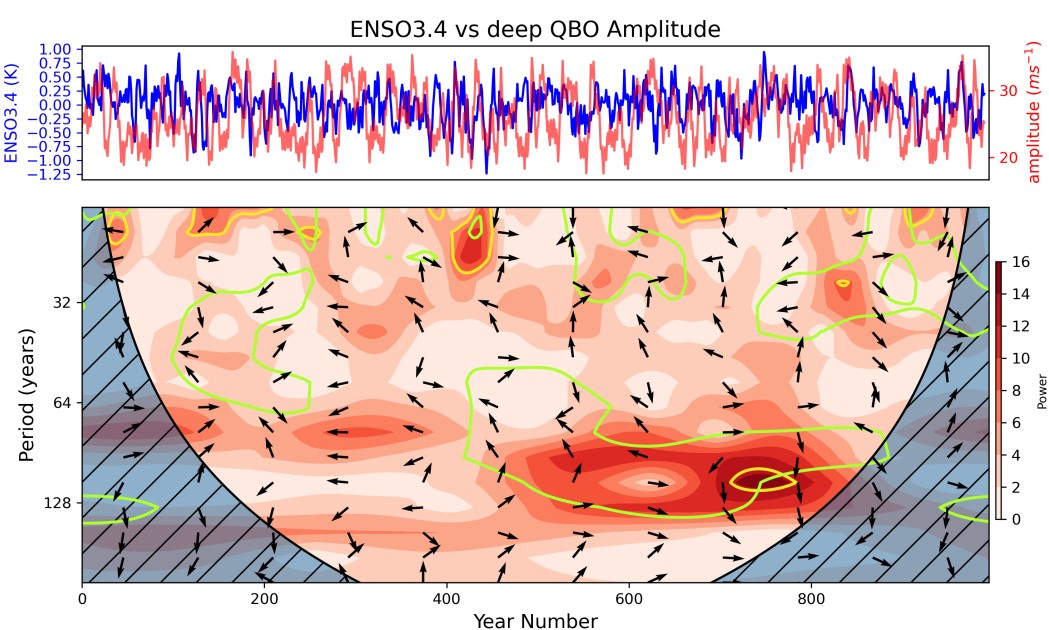

**Figure A5. Top**: Sep-Nov ENSO3.4 index mean (blue) and Sep-Nov deep QBO Hilbert Amplitude index smoothed with a 5 year window (red). **Bottom**: Cross wavelet power spectrum between the two time series.





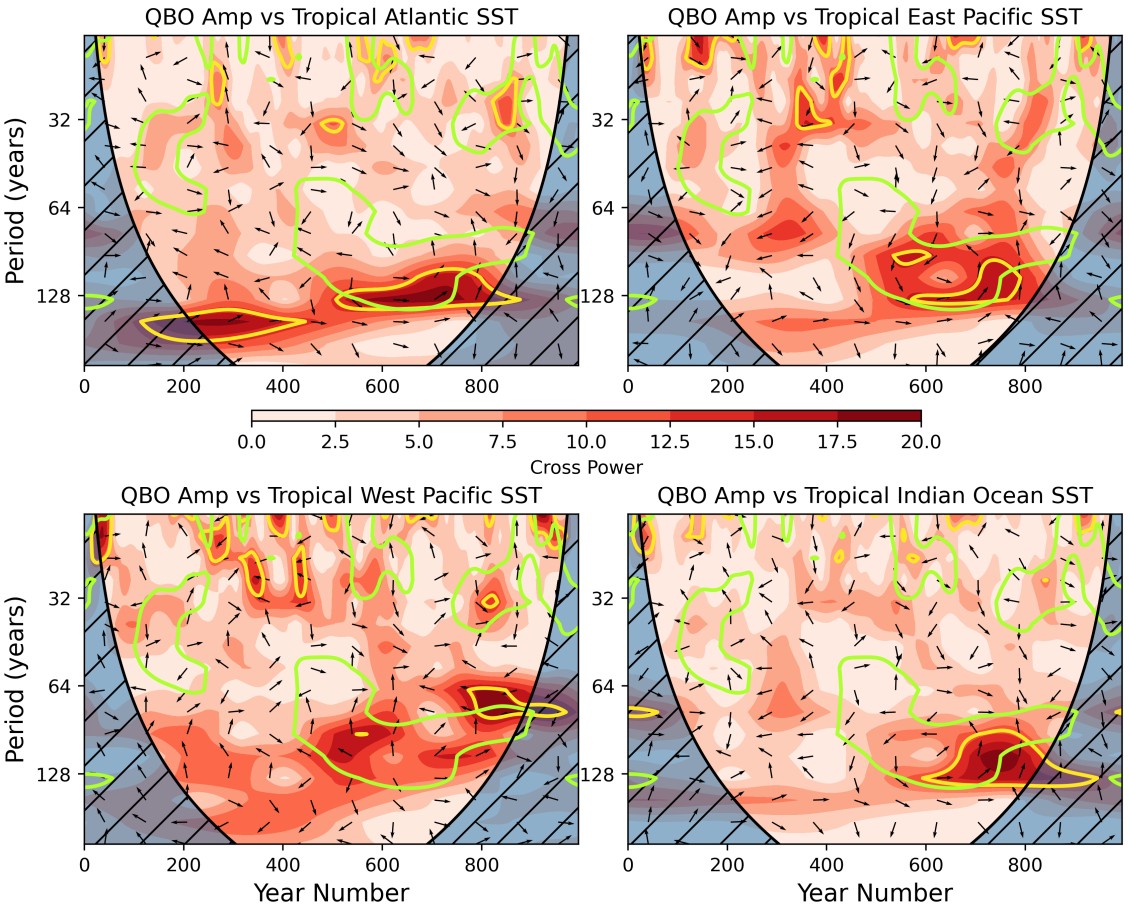

**Figure A6.** Cross wavelet power spectra between Sep-Nov deep QBO amplitude modulation and Sep-Nov SST anomaly in each Tropical basin.