# Peer review of "Origins of Multi-decadal Variability in Sudden Stratospheric Warmings"

_Weather and Climate Dynamics, 2020_

## Referee Comment (RC2) · Anonymous Referee #2 · 7 Dec 2020

General Comments:

In this new study, the authors using a CMIP6 pre-industrial control run from the UKESM global climate model (GCM) to its evaluate internal variability in sudden stratospheric warming (SSW) events. While there is some limited evidence of SSW multi-decadal variability in observations, the atmospheric reanalysis record is too short to completely address this issue (e.g., internally versus externally-driven). Here, the authors use several transformation methods of wavelet time series analysis to investigate SSW variability in the control simulation and the physical sources that may be associated with it. In particular, the wavelet analysis reveals an important connection between the (deep) westerly phase of the Quasi-Biennial Oscillation (QBO) and multi-decadal periods of little to no SSW activity. In agreement with earlier studies, the vertical structure of the

QBO is important for addressing the Holton-Tan effect, and thus, also SSW variability.

Overall, this is a very interesting study that addresses an open issue of multi-decadal variability in stratosphere-troposphere interactions, which will be of significant interest to the weather and climate communities. In particular, there have been few attempts to assess SSW variability using a GCM with a well-resolved stratosphere and a realistic internal QBO. However, I think a few more caveats should be more explicitly mentioned given that models still struggle to simulate dynamical coupling between the stratosphere and troposphere. Further, this study is only using one model (UKESM). While I am not overly familiar with some of the wavelet transformations employed in this study, more caution (or additional analysis) is needed for interpreting the potential sources of SSW multi-decadal variability (e.g., tropical SSTs – ENSO) that are investigated here.

Recommendation: The paper will be acceptable for publication in Weather and Climate Dynamics after some major revisions.

Specific Comments:

1. L10-11; Do you think they account for some SSW variability or could it just be coincidental internal noise?

2. L20-21; Reference Domeisen et al. (2020) for importance of SSW to S2S forecasts

3. L25-26; Cohen et al. (2009) investigated changes in wave activity/surface forcing on stratospheric variability

4. L29-33; Seviour (2017) attributed the recent weakening of the polar vortex to internal variability

5. L91-95; Reword this sentence to improve clarity

6. L96-101; A very brief discussion would be helpful here to mention other surface forcings that may modulate the strength (and perhaps decadal variability) of the stratospheric polar vortex in observation records (Garfinkel et al. 2010). Moreover, recent studies have found that boundary conditions, such as sea ice and snow cover, may modulate the Holton-Tan relationship or even QBO cycle (Hirota et al. 2018, Labe et al. 2019). For example, SSW variability may occur through enhanced vertical wave activity due to Arctic sea ice loss (e.g., Kim et al. 2014; Nakamura et al. 2016) and/or Eurasian snow cover anomalies (e.g., Cohen et al. 2007; Henderson et al. 2018).

7. L110; Why is this unexpected? The vertical structure of the QBO has been identified in numerous studies for its importance to variability of stratosphere-troposphere coupling. Perhaps reword to improve reader clarity.

8. L130-131; Change to something like: "To compare the climate model with the recent observational record, we use ERA-Interim reanalysis (Dee et al. 2011)."

9. L204; How sensitive are the results to your choice of SSW definition?

10. L213-214; Restate the definition of the ENSO3.4 index here.

11. L214-216; Why is this metric chosen as a proxy for the Aleutian Low, instead of something simpler like the central pressure as in Overland et al. (1999)? Reference?

12. L228-229; In my view, there looks to be a statistically significant difference in the number of SSW events distributed per month in Figure 1.

13. L230-240; Although a comparison between model and reanalysis is great, it should still be noted that the samples are not completely comparable if SSWs are influenced by external forcing (climate change) in the real world.

14. L239-246; Again, additional caveats about the use of one model for this analysis are needed... i.e., difference in QBO period (common in high-top models), which could affect the overall conclusions.

15. L321-322; This 90-year periodicity looks somewhat large though in Figure 8? Is there something physically-related to this or is it internal noise? Have you done any

lead-lag or regression composites to further investigate any ENSO-SSW relationship at this time-scale?

16. L360; It could also be just noise in the short reanalysis record.

17. L366-367; This is difficult to see in Figure 10. Could the left six panels be modified slightly?

18. L375-378; This conclusion seems particularly sensitive to the QBO definition. Any thoughts?

19. L409-412; Where is this shown?

20. L416-418; Reword sentence to improve clarity.

Technical Comments:

1. L6; "... coupled Atmosphere-Ocean-Land-Sea ice model." to "coupled global climate model."

2. L46; "...and the [stratospheric polar] vortex."

3. L49; "link" to "effect"

4. L74 and throughout; Unless you are talking about the vertical structure of the Aleutian Low, change "depth" to something like "strength" or "intensity"

5. L91; Lowercase "tropical"

References:

Cohen, J., Barlow, M., Kushner, P. J., & Saito, K. (2007). Stratosphere–troposphere coupling and links with Eurasian land surface variability. Journal of Climate, 20(21), 5335-5343.

Cohen, J., Barlow, M., & Saito, K. (2009). Decadal fluctuations in planetary wave forcing modulate global warming in late boreal winter. Journal of Climate, 22(16), 4418-

4426.

Domeisen, D. I., Butler, A. H., Charlton‐Perez, A. J., Ayarzagüena, B., Baldwin, M. P., Dunn‐Sigouin, E., ... & Kim, H. (2020). The role of the stratosphere in subseasonal to seasonal prediction: 2. Predictability arising from stratosphere‐troposphere coupling. Journal of Geophysical Research: Atmospheres, 125(2), e2019JD030923.

Garfinkel, C. I., Hartmann, D. L., & Sassi, F. (2010). Tropospheric precursors of anomalous Northern Hemisphere stratospheric polar vortices. Journal of Climate, 23(12), 3282-3299.

Henderson, G. R., Peings, Y., Furtado, J. C., & Kushner, P. J. (2018). Snow–atmosphere coupling in the Northern Hemisphere. Nature Climate Change, 8(11), 954-963.

Hirota, N., Shiogama, H., Akiyoshi, H., Ogura, T., Takahashi, M., Kawatani, Y., ... & Mori, M. (2018). The influences of El Nino and Arctic sea-ice on the QBO disruption in February 2016. npj Climate and Atmospheric Science, 1(1), 1-5.

Kim, B. M., Son, S. W., Min, S. K., Jeong, J. H., Kim, S. J., Zhang, X., ... & Yoon, J. H. (2014). Weakening of the stratospheric polar vortex by Arctic sea-ice loss. Nature communications, 5(1), 1-8.

Labe, Z., Peings, Y., & Magnusdottir, G. (2019). The effect of QBO phase on the atmospheric response to projected Arctic sea ice loss in early winter. Geophysical Research Letters, 46(13), 7663-7671.

Nakamura, T., Yamazaki, K., Iwamoto, K., Honda, M., Miyoshi, Y., Ogawa, Y., ... & Ukita, J. (2016). The stratospheric pathway for Arctic impacts on midlatitude climate. Geophysical Research Letters, 43(7), 3494-3501.

Overland, J. E., Adams, J. M., & Bond, N. A. (1999). Decadal variability of the Aleutian low and its relation to high-latitude circulation. Journal of climate, 12(5), 1542-1548.

Seviour, W. J. (2017). Weakening and shift of the Arctic stratospheric polar vortex: Internal variability or forced response?. Geophysical Research Letters, 44(7), 3365-3373.

---

## Referee Comment (RC3) · Anonymous Referee #3 · 21 Dec 2020

The present manuscript analyzes the sources of multi-decadal variability in major Sudden Stratospheric Warmings (SSWs). To do so, the authors apply a wavelet spectral decomposition method to the output of a 1000-yr pre-industrial control simulation of the UKESM model. The results reveal a periodicity of SSWs variability of approximately 60-90 years during 450 years of the simulation. Among the studied phenomena, long-term variability of the (deep) QBO amplitude seems to be the most important. In contrast, variability in tropical sea surface temperatures and Aleutian low do not seem relevant to explain long-term SSW variability.

The analysis of multidecadal variability is a very interesting topic that is recently receiving increasing attention. However, only a few papers have addressed it due to the unavailability of long observational data record and the relatively low number of

very long model simulations with daily output. Thus, a 1000-yr pre-industrial control simulation as that used in this paper provides a good opportunity to perform such a type of analysis. The findings identified in this study, in particular, the connection between long-term variability of QBO and SSWs, are certainly promising. However, they are mainly focused on the results provided by the wavelet analysis, and I miss an attempt to explain the detected relationships. Based on this and my following specific comments, I think there are major corrections that the authors should address before recommending its publication.

Specific comments:

L21: There are earlier papers that relate the occurrence of SSWs to cold air outbreaks in mid-latitudes such as Tomassini et al. (2012) or Lehtonen and Karpechko (2016).

L130-131: The historical simulation might be better for model validation than the pre-industrial one. Changes in GHGs concentrations may affect the distribution of SSWs.

L214-216 and L339: In the first sentences the authors indicate that the Aleutian low index is computed for September-November, but along Section 3.4 it is written that it is computed for December-March. From my point of view the later would be more accurate, as it will be simultaneous to the months considered for the occurrence of SSWs.

L221-229: I am surprised that the frequency of SSWs in November does not appear in the figure. Moreover, the sum of the monthly frequency of SSWs for December-March in the model does not give 0.54 events/winter but approximately 0.1 events/winter less. Is it possible that November shows a frequency of around 0.1 SSWs/winter? If so, that means that UKESM is one of those models that presents a too weak vortex in November and an artificially high frequency of SSWs in that month. I was wondering if this is the reason why the authors restrict the analysis to the period between December-March. I would agree that SSWs in November are unrealistic but I have some concerns about not considering them. The occurrence of SSWs in November

will precondition the state of the vortex in December-January as it will be recovering and probably anomalously strong. In that case, it will prevent the occurrence of an SSW. This might distort the results for the whole winter. I think it might be worthy to repeat the analysis considering the November SSWs to check if conclusions remain the same.

L 315-318: The mature phase of El Niño events is reached from November to January (Wang 2002). I agree with the authors that it will likely remain in the same state between early and mid-winter but I think it will be important to check it.

L318-321: Some authors have related PDO and ENSO (Verdon and Franks, 2006). Is the low frequency of ENSO related to the PDO in this case?

L337-353: I was wondering if the lack of correlation between the variability of Aleutian Low and SSW can be explained by the fact that the region selected for the Aleutian Low does not coincide with that associated with precursors of SSWs. In this sense, Garfinkel et al (2012) investigate the reason why the SSW frequency in El Niño and La Niña winters is similar in observations. The find that both La Niña and El Niño lead to circulation anomalies of the same sign in the area associated with SSW precursors. I think it would be important to identify the areas of precursors of SSWs following Garfinkel et al. (2012) or Garfinkel et al (2010) and compare with the spatial pattern of the Aleutian Low that the authors compute.

Technical comments: L44: north –> North In some figures such as Figure 11 or 5, the authors use lower case a), b) and so on to refer to the different panels of a figure and in other figures such as 8 or 9 the authors use upper case A) and B) for the same purpose.

L404: figure 12 b

L409: please include (not shown) at the end of the sentence.

References: Garfinkel, C. I., D. L. Hartmann and F. Sassi, 2010: Tropospheric precursors of anomalous Northern Hemisphere stratospheric polar vortices. J. Climate, 23, 3282-3299.

Garfinkel, C. I, A. H. Butler, D. W. Waugh, M. M. Hurwitz and L. M. Polvani, 2012: Why might stratospheric sudden warmings occur with similar frequency in El Niño and La Niña winters? J. Geophys. Res., 117, D19106, doi:10.1029/2012JD017777.

Lehtonen, I. and A. Y. Karpechko, 2016: Observed and modeled tropospheric cold anomalies associated with sudden stratospheric warmings, J. Geophys. Res. Atm., 121, 1591-1610, doi:10.1002/2015JD023860

Tomassini, L., E. P. Gerber, M. P. Baldwin, F. Bunzel and M. Giorgetta, 2012: The role of strastosphere-troposphere coupling in the occurrence of extreme winter cold spells over northern Europe, J. Adv. Mod. Ear. Sys., 4, M00A03. doi:10.1029/2012MS000177.

Verdon, D. C and S. Franks, 2006: Long-term behavior of ENSO: Interactions with the PDO over the past 400 years inferred from paleoclimate records, Geophys. Res. Lett.,33, L06712, doi:10.1029/2005GL025052.

Waugh, C., 2002: Atmospheric circulation cells associated with the El Niño-Southern Oscillation, J. Climate, 15, 399-419.

---

## Author Comment (AC1) · 23 Dec 2020

**Response to Reviewer Comments**

**"Origins of Multi-decadal Variability in Sudden Stratospheric Warmings" by Oscar Dimdore-Miles et al.**

We thank all the reviewers for providing their comments on our analysis. Their questions and suggestions have helped us to consider the role of ENSO and the PDO in multi-decadal SSW signals more closely as well as make our description of our wavelet methodology and the interpretation of wavelet plots clearer to the intended reader.

**Summary of major changes**

- Additional analysis using multi-linear techniques (new section 3.2) to explore the comparative roles of ENSO, AL and QBO forcing (Tables 1-3) ; the results support the wavelet results and strengthen our conclusions on the role of the QBO amplitude modulation.
- Section 2 (previously 'Model and Data') renamed 'Methodology' to better reflect its content; now includes an improved description and justification of the wavelet technique;.
- All wavelet analysis figures have been replotted using consistent colour scales for easier comparisons.
- Better justification of which months have been analysed, including an extra supporting figure showing results for NDJFM.
- Improved discussion of the potential role of the PDO and an additional supporting figure.

*The authors analyze the possible causes that lead to extended periods with and without Stratospheric sudden warmings. They find that such events vary on multi-decadal timescales of period between 60 and 90 years, and that signals on these timescales are present for approximately 450 years of a 1000-year long simulation. While tropical sea surface temperatures and Aleutian Low variability seem relatively unimportant, the amplitude of the stratospheric quasi biennial oscillation (QBO) westerlies in the mid stratosphere between 15hPa and 30hPa seems more important. This paper has some interesting results, and it is very likely that the paper will be suitable for publication after revision. However I must admit that I am not an expert in the methods the authors use, and I had some questions concerning the manner in which the authors draw conclusions from their results.*

*Major comments: 1. This reviewer has limited experience with the chief methodology used in this paper, and I suspect that most of the intended audience has a similar lack of familiarity. While advanced methods can help uncover relationships that would otherwise be missed, the interpretability of the resulting effect is often missing. While I appreciate the effort the authors put it to interpreting the results of the wavelet analysis, there are some (perhaps very basic) questions I had.*

We agree that many will be less familiar with wavelet analysis compared with multi-linear regression. To address this, we have added a new section (3.2) where we show some multi-linear regression analysis (Tables 1) and discuss its limitations, as a lead-in to the section on wavelet analysis. We have also added further regression analysis (Tables 2-3) that provide

supporting evidence for the wavelet analysis. Additional text in our introduction and at the end of this new section 3.2 further motivates the use of wavelet methods.

*At present the arguments concerning the relative importance of ENSO and the QBO are hand-wavy. The authors claim near line 325 that the ENSO signal is weak in Figure 8. However by eye, the ENSO signal in Figure 8 near the 90 year periodicity is actually little different from the QBO signal in Figure 12. There is no colorbar on either figure, but the shade of red shown is similar, and is located in a similar location. Maybe if I squint I can see the authors point, but there must be a better way to quantify the relatively importance. Is there a way to compute some sort of different plot between figure 12 and figure 8 to more robustly make the point?*

Apologies for this lack of clarity - each spectra was normalised by the variance of the time series which was not clear from our text (this is now rectified in section 2.3), and our interpretation was based primarily on the region of statistical significance. We have clarified the text and improved the figures so that all wavelet spectra use the same shading levels (with the colorbar added). It is now much more evident that ENSO exhibits significantly weaker power at the relevant periods compared to the QBO metric.

*The simplest way to evaluate this point using more classical techniques is with simple regression. You compute the regression coefficient (including uncertainty) between the QBO signal and SSW_5yr, and between ENSO and SSW_5yr, and simply compare the regression coefficients.*

As described above, we now include multilinear regression analyses comparing the contributions of ENSO, the AL and the deep QBO amplitude to SSW_5yr – see the new tables 1-3 and accompanying discussion.

*What exactly is the ENSO signal at a periodicity of 90 years? In addition to the modes of variability considered by the authors, there is an additional oceanic mode: the PDO or IPO (Pacific decadal oscillation or Interdecadal Oscillation). The periodicity in observations is a bit shorter than 90 years, but perhaps in the authors' model the periodicity is longer. The PDO and IPO project strongly onto ENSO, and indeed one leading forcing of the PDO is simply a low-passed version of ENSO (Newman et al 2016). The PDO has been linked to vortex variability (Rao et al 2019, but see the papers cited therein). Please clarify whether the link seen in Figure 8 is just the PDO.*

Thank you for raising this possibility - the PDO was not discussed sufficiently in our analysis. We have added text to note the possibility that the ENSO signal at 90 years may be a manifestation of the PDO (lines 398-401). We have added a supporting figure (A3) to discuss its importance. However, while we agree that it merits further investigation, we believe this to be outside the remit of the current study, given that it does not account for a large portion of SSW_5yr power in our model (and not as much as the QBO metrics can).

*I found the QBO index the authors choose a bit strange. The strongest HT relationship on interannual timescales is with the QBO near 40hPa or 50hPa and the vortex. If one focuses on winds near 20hPa, the HT relationship more or less goes away entirely on interannual timescales. The authors claim that on longer timescales, the most important QBO phase is winds between 15hPa and 30hPa, not winds lower in the stratosphere. While it is possibly conceivable that the phase on short and long timescales isn't identical, this much of a mismatch in phases is disturbing, at least to this reviewer. Do the authors have any ideas on*

*possible causes for such a shift in phase? This mismatch casts some doubt on the robustness of the results, as at present the search for the "best" phase seems too much like a data-fishing exercise.*

Our rationale for focusing on the deep QBO metric follows from the work of Gray et al. (2018) and Andrews et al. (2019) who demonstrated a stronger surface response to the QBO using this metric compared to single levels (and ultimately, we are interested to understand not only the source of long-term vortex variability but also long-term surface variability). We have added some additional text to section 3.1 to note that there is no reason why the deep QBO composite should look like the average of the corresponding single-level composites. We do not believe that the lack of a clear HT response to the single-level 20 hPa QBO is a problem: let us assume that a deep region of QBO easterlies between 15-30 hPa is important for development of an SSW and hence weakened vortex. Selecting years simply based on the 20 hPa will indeed select those years with a deep easterly QBO (and these years will indicate a weakened vortex) but it will also select years with strong vertical shear above and/or below that level, which is likely to obscure the vortex response (since our assumption is that a weak vortex response requires the absence of vertical shear), thus explaining the null response to this single level. (We also note that even though there is no strong HT signal from the 20 hPa level, the 20 hPa timeseries exhibits substantial amplitude variability in the westerly phase, hence our use of the Hilbert amplitude).

***Minor comments:***
*Line 21: there are earlier studies showing an impact of SSW on cold snaps. See Thompson et al 2002 and Lehtonen and Karpechko 2016.*
Both references added.

*Line 32: Using a subset of the simulations from Garfinkel et al 2017, Garfinkel et al 2015 actually found a slight strengthening of the vortex in late winter due to SSTs over the 30-year period from 1980 to 2009.*
Reference to this work and the possible role of SST forcing of the vortex on decadal timescales included (lines 35-37).

*Line 52: Garfinkel et al 2018 also showed results regarding the HT effect in GloSea5, which is based on HadGEM3 GC2.0 if I understand the S2S website correctly. The model performed well as compared to reanalysis.*
Thank you, reference added (line 57).

*Line 75: The importance of the Aleutian low for stratospheric wave driving was shown earlier than 2015. See Manzini et al 2006, Taguchi and Hartmann 2006, and Garfinkel et al 2010 The former two studies didn't focus on the Aleutian Low per se, but show this effect.*
References added (line 96)

*Line 93-94: A competition between Pacific and Indian Ocean SSTs for El Nino was shown earlier by Fletcher and Kushner*
References added (line 100)

*Line 214-215 Is the Aleutian index defined for autumn only?*

This was a typo and should read Dec-Mar as this is the month range used in the figures. We tested a range of Autumn and Winter month ranges and the spectra are relatively robust across the season on the timescales we consider (60-90 years). Dec-Mar covers the same months used to define SSWs and was a natural choice to use encompassing mid-late winter. We have added a statement of this testing of different month ranges (lines 253-256).

*Line 339 states you used DJFM. Please clarify*
Mention of Sep-Nov on lines 214-215 is incorrect - Dec-Mar was used throughout – corrected.

*Line 229: Actually the seasonal evolution of SSWs doesn't match observations all that well (non-overlapping error bars), with too many SSWs later in winter as compared to DJF. Such a model bias is fairly common however, and Horan and Reichler 2017 argue that the observed seasonal distribution may reflect sampling uncertainty.*
Additional discussion of the seasonal evolution has been added, including November SSWs, and reference included (lines 270-277).

*Caption of Figure A4 and the figure itself don't seem to match. Please correct. For now, the claim near line 352-353 stands unsupported.*
The caption has been rewritten and the statement on 431 in the revised manuscript is now supported.

---

## Author Comment (AC3) · 23 Dec 2020

**Response to Reviewer Comments**

**"Origins of Multi-decadal Variability in Sudden Stratospheric Warmings" by Oscar Dimdore-Miles et al.**

We thank all the reviewers for providing their comments on our analysis. Their questions and suggestions have helped us to consider the role of ENSO and the PDO in multi-decadal SSW signals more closely as well as make our description of our wavelet methodology and the interpretation of wavelet plots clearer to the intended reader.

**Summary of major changes**

- Additional analysis using multi-linear techniques (new section 3.2) to explore the comparative roles of ENSO, AL and QBO forcing (Tables 1-3); the results support the wavelet results and strengthen our conclusions on the role of the QBO amplitude modulation.
- Section 2 (previously 'Model and Data') renamed 'Methodology' to better reflect its content; now includes an improved description and justification of the wavelet technique;.
- All wavelet analysis figures have been replotted using consistent colour scales for easier comparisons.
- Better justification of which months have been analysed, including an extra supporting figure showing results for NDJFM.
- Improved discussion of the potential role of the PDO and an additional supporting figure.

The present manuscript analyzes the sources of multi-decadal variability in major Sudden Stratospheric Warmings (SSWs). To do so, the authors apply a wavelet spectral decomposition method to the output of a 1000-yr pre-industrial control simulation of the UKESM model. The results reveal a periodicity of SSWs variability of approximately 60-90 years during 450 years of the simulation. Among the studied phenomena, long-term variability of the (deep) QBO amplitude seems to be the most important. In contrast, variability in tropical sea surface temperatures and Aleutian low do not seem relevant to explain long-term SSW variability. The analysis of multidecadal variability is a very interesting topic that is recently receiving increasing attention. However, only a few papers have addressed it due to the unavailability of long observational data record and the relatively low number very long model simulations with daily output. Thus, a 1000-yr preindustrial control simulation as that used in this paper provides a good opportunity to perform such a type of analysis. The findings identified in this study, in particular, the connection between long-term variability of QBO and SSWs, are certainly promising. However, they are mainly focused on the results provided by the wavelet analysis, and I miss an attempt to explain the detected relationships. Based on this and my following specific comments, I think there are major corrections that the authors should address before recommending its publication.

Thank you for your supportive comments. We agree that the majority of our results are derived from the wavelet spectra analysis. We have added additional text explaining why we believe wavelet analysis is the best technique to use, and we have also added a new section (3.2) where we show results from a multi-linear regression analysis (Table 1) and discuss its limitations (as a lead-in to our wavelet analysis). We also present some further regression

analysis (Tables 2-3) later in the results section that provide supporting evidence for the wavelet analysis results. We hope the reviewer agrees that this provides more balance to our study, as well as strengthening the conclusions.

We do not believe it is possible to explain the detected relationships without substantially more research (and we would prefer not to speculate at this stage). At this stage of the research, we report on a comprehensive investigation of the relationships that we expected to see (primarily from the surface in terms of SST and associated variability) and found them to be inadequate. We then explored alternative relationships that involved the QBO, in particular an amplitude modulation of the QBO that had not previously been discussed to our knowledge. We are planning a follow-on paper where we will explore the origins of this QBO amplitude modulation (which may still be related to SSTs via equatorial wave forcing – we are not sure) but the current paper is already overly long. Similarly, explaining the relationship between the amplitude modulation of the deep QBO index and the strength of the polar vortex is also a challenging task which will require targeted model experiments to understand the wave mean flow interactions. We have some experiments underway that will be reported separately.

L21: There are earlier papers that relate the occurrence of SSWs to cold air outbreaks in mid-latitudes such as Tomassini et al. (2012) or Lehtonen and Karpechko (2016). References added.

**L130-131: The historical simulation might be better for model validation than the preindustrial one. Changes in GHGs concentrations may affect the distribution of SSWs.**

We have added a caveat stating the possibility of an anthropogenic warming signal in the ERA-Interim warming rates (lines 270-277). We also refer to Andrews et al. 2020 for validation of the historical simulation of the model and agree that consideration of such work is good practice.

We caveat this however, with the fact that the nature of climate change signal in SSWs is not well understood (and varies significantly across different models) as outlined in Ayarzagüena et al. 2020.

**L214-216 and L339: In the first sentences the authors indicate that the Aleutian low index is computed for September-November, but along Section 3.4 it is written that it is computed for December-March. From my point of view the later would be more accurate, as it will be simultaneous to the months considered for the occurrence of SSWs.**

The statement in the first sentences was a typo and should read Dec-Mar as this is the month range used in the figures. We tested a range of Autumn and Winter month ranges and the spectra are relatively robust across the season on the timescales we consider (60-90 years). As the reviewer points out, Dec-Mar covers the same months used to define SSWs and was a natural choice to use encompassing mid-late winter. We have added a statement of this testing of different month ranges on lines 254-255.

L221-229: I am surprised that the frequency of SSWs in November does not appear in the figure. Moreover, the sum of the monthly frequency of SSWs for December-March in the model does not give 0.54 events/winter but approximately 0.1 events/winter less.

Is it possible that November shows a frequency of around 0.1 SSWs/winter? If so, that means that UKESM is one of those models that presents a too weak vortex in November and an artificially high frequency of SSWs in that month. I was wondering if this is the reason why the authors restrict the analysis to the period between December-March. I would agree that SSWs in November are unrealistic but I have some concerns about not considering them. The occurrence of SSWs in November will precondition the state of the vortex in December-January as it will be recovering and probably anomalously strong. In that case, it will prevent the occurrence of an SSW. This might distort the results for the whole winter. I think it might be worthy to repeat the analysis considering the November SSWs to check if conclusions remain the same.

This is a fair point and the reviewer is correct in suggesting that UKESM has an unusually high SSW rate in November and that this was the reason for omitting them from the analysis – an artificially high number of November warmings may have obscured the QBO, ENSO and AL signal in SSW\_5yr if the origin of this bias was elsewhere (such as due to early formation of the vortex as suggested in Mennary et al. 2018 which analyses HadGEM3GC3.1, a sister model of UKESM1). we have discussed the November bias in the text on lines 275-277 and have carried out checks to ensure that excluding November does not significantly influence our main conclusions; in view of this we decided that it was not necessary to amend the figure

We have calculated the spectra for SSW\_5yr which includes November warmings and included it as supporting figure A1. It shares the key features of the spectrum presented in the paper – persistent power at 60-90 years for ~450 years of the simulation.

**L 315-318: The mature phase of El Niño events is reached from November to January (Wang 2002). I agree with the authors that it will likely remain in the same state between early and mid-winter but I think it will be important to check it.**

We agree with the reviewer that this is an important check. When we evaluate ENSO3.4 index in Nov-Jan (as Wang (2002) recommend), we obtain an index that correlates highly with the corresponding ENSO3.4 evaluated in Sep-Nov (r = 0.94). Furthermore, this correlation applies on multi-decadal timescales shows by the correlation between the 2 ENSO indices after they have been fourier filtered to retain only power corresponding to periods greater than 60 years (r = 0.92). This suggests our results regarding ENSO are robust to the month range chosen. We have added text in the methodology section to say that sensitivity checks were performed to ensure that results were not sensitive to the choice of months (line 255).

**L318-321: Some authors have related PDO and ENSO (Verdon and Franks, 2006). Is the low frequency of ENSO related to the PDO in this case?**

Thank you for raising this possibility - the PDO was not discussed sufficiently in our analysis. We have added text (within section 3.4, lines 398-400) to note the possibility that the ENSO signal at 90 years may be a manifestation of the PDO. We have added supporting figure A3 to discuss its importance. However, while we agree that it merits further investigation, we believe this to be outside the remit of the current study, given that it does not account for a large portion of SSW\_5yr power (and not as much as the QBO metrics can).

L337-353: I was wondering if the lack of correlation between the variability of Aleutian

Low and SSW can be explained by the fact that the region selected for the Aleutian Low does not coincide with that associated with precursors of SSWs. In this sense, Garfinkel et al (2012) investigate the reason why the SSW frequency in El Niño and La Niña winters is similar in observations. The find that both La Niña and El Niño lead to circulation anomalies of the same sign in the area associated with SSW precursors. I think it would be important to identify the areas of precursors of SSWs following Garfinkel et al. (2012) or Garfinkel et al (2010) and compare with the spatial pattern of the Aleutian Low that the authors compute.

This is another fair point as we have used a PC based method to pick up the variability of the AL as opposed to a box in Garfinkel et al. (2012) which is associated with SSWs. The box suggested by Garfinkel et al. (2012) (52.5°N–72.5°N, 165°E–195°E) overlaps slightly but lies slightly further north of the centre of the 1st EOF of SLP so merits some analysis to check if this may be a better metric to use.

An index defined as the area weighted average de-seasonalised SLP anomaly within the box exhibits a significant but somewhat small correlation with our existing metric (r = 0.4, p < 0.01) suggesting there may be some variability missed by our metric. However, the average over this box exhibits lower correlation with our SSW time series than the existing AL index: r = -0.21 for our AL and r = -0.13 for the box-based AL. This suggests we are not missing a region of high influence over the vortex with our existing measure.

Technical comments: L44: north -> North Changed to North.

In some figures such as Figure 11 or 5, the authors use lower case a), b) and so on to refer to the different panels of a figure and in other figures such as 8 or 9 the authors use upper case A) and B) for the same purpose.

Changed all subfigure labels to lower case.

L404: figure 12 b

Remains unchanged (see previous comment).

L409: please include (not shown) at the end of the sentence. (not shown) added.

---

## Author Comment (AC2)

**Response to Reviewer Comments**

**"Origins of Multi-decadal Variability in Sudden Stratospheric Warmings" by Oscar Dimdore-Miles et al.**

We thank all the reviewers for providing their comments on our analysis. Their questions and suggestions have helped us to consider the role of ENSO and the PDO in multi-decadal SSW signals more closely as well as make our description of our wavelet methodology and the interpretation of wavelet plots clearer to the intended reader.

**Summary of major changes**

- Additional analysis using multi-linear techniques (new section 3.2) to explore the comparative roles of ENSO, AL and QBO forcing (Tables 1-3) ; the results support the wavelet results and strengthen our conclusions on the role of the QBO amplitude modulation.
- Section 2 (previously 'Model and Data') renamed 'Methodology' to better reflect its content; now includes an improved description and justification of the wavelet technique;.
- All wavelet analysis figures have been replotted using consistent colour scales for easier comparisons.
- Better justification of which months have been analysed, including an extra supporting figure showing results for NDJFM.
- Improved discussion of the potential role of the PDO and an additional supporting figure.

*General Comments: this new study, the authors using a CMIP6 pre-industrial control run from the UKESM global climate model (GCM) to its evaluate internal variability in sudden stratospheric warming (SSW) events. While there is some limited evidence of SSW multi-decadal variability in observations, the atmospheric reanalysis record is too short to completely address this issue (e.g., internally versus externally-driven). Here, the authors use several transformation methods of wavelet time series analysis to investigate SSW variability in the control simulation and the physical sources that may be associated with it. In particular, the wavelet analysis reveals an important connection between the (deep) westerly phase of the Quasi-Biennial Oscillation (QBO) and multi-decadal periods of little to no SSW activity. In agreement with earlier studies, the vertical structure of the QBO is important for addressing the Holton-Tan effect, and thus, also SSW variability.*

*Overall, this is a very interesting study that addresses an open issue of multi-decadal variability in stratosphere-troposphere interactions, which will be of significant interest to the weather and climate communities. In particular, there have been few attempts to assess SSW variability using a GCM with a well-resolved stratosphere and a realistic internal QBO. However, I think a few more caveats should be more explicitly mentioned given that models still struggle to simulate dynamical coupling between the stratosphere and troposphere. Further, this study is only using one model (UKESM).*
*Recommendation: The paper will be acceptable for publication in Weather and Climate Dynamics after some major revisions.*

Thank you for your supportive comments. We have added some text to the final section outlining the caveats noted (lines 540-547), and also where appropriate throughout the text.

*While I am not overly familiar with some of the wavelet transformations employed in this study, more caution (or additional analysis) is needed for interpreting the potential sources of SSW multi-decadal variability (e.g., tropical SSTs – ENSO) that are investigated here.*
We have added some substantial new analysis using multi-linear regression (please also see response to a similar issue raised by reviewer 1) including three new tables (tables 1-3) of results as a lead-in to our wavelet analysis, with added text discussing the relative merits of each type of analysis, together with improved description of wavelet analysis and what it can tell us. The new analysis supports our interpretation that while there is some contribution from long-term variability of the ENSO (and PDO, which is also now discussed) there is a stronger link to the deep QBO.

*Specific Comments*
*1.L10-11; Do you think they account for some SSW variability or could it just be coincidental internal noise?*
The Aleutian low exhibits coincident power for approximately 100 years which is probably not just internal random noise, but the selected SST regions (particularly tropical west pacific) shows significantly less coincident power with SSWs and is more likely coincidental internal noise. The message we wanted to convey with this statement is that the surface indices we examined cannot sufficiently account for the SSW variability over the whole ~400 years which is why we look elsewhere (namely the QBO region) for additional sources of multi-decadal signals.

*2. L20-21; Reference Domeisen et al. (2020) for importance of SSW to S2S forecasts*
Reference added

*3. L25-26; Cohen et al. (2009) investigated changes in wave activity/surface forcing on stratospheric variability*
Reference added

*4. L29-33; Seviour (2017) attributed the recent weakening of the polar vortex to internal Variability*
Reference added

*5. L91-95; Reword this sentence to improve clarity*
We have split up this sentence and reworded this to
*"SSTs in other tropical regions also exhibit coherence with the vortex. Rao and Ren (2017) show that Tropical Atlantic SSTs give rise to a vortex response although 100 it is highly variable throughout the season while Fletcher and Kushner (2011), Fletcher and Kushner (2013) and Rao and Ren (2015) propose a tropical Indian Ocean (TIO) connection. Positive TIO SST anomalies lead to a reduced strength of the AL that weakens the Rossby wave forcing of the vortex, an opposite effect to the ENSO-vortex connection where positive SST anomalies leads to vortex weakening."*

*6. L96-101; A very brief discussion would be helpful here to mention other surface*

*forcings that may modulate the strength (and perhaps decadal variability) of the stratospheric polar vortex in observation records (Garfinkel et al. 2010). Moreover, recent studies have found that boundary conditions, such as sea ice and snow cover, may modulate the Holton-Tan relationship or even QBO cycle (Hirota et al. 2018, Labe et al. 2019). For example, SSW variability may occur through enhanced vertical wave activity due to Arctic sea ice loss (e.g., Kim et al. 2014; Nakamura et al. 2016) and/or Eurasian snow cover anomalies (e.g., Cohen et al. 2007; Henderson et al. 2018).*

An additional paragraph has been added (lines 104-113), outlining the possible role of these different surface forcings.

*7. L110; Why is this unexpected? The vertical structure of the QBO has been identified in numerous studies for its importance to variability of stratosphere-troposphere coupling. Perhaps reword to improve reader clarity.*

We were referring to the amplitude modulation of a deep QBO, which to our knowledge has not been discussed previously (line 128-129). We have reworded to make this clearer.

*8. L130-131; Change to something like: "To compare the climate model with the recent observational record, we use ERA-Interim reanalysis (Dee et al. 2011)."*

We have changed this as recommended.

*9. L204; How sensitive are the results to your choice of SSW definition?*

The number of identified SSWs does not change substantially if we use slightly different variants on the definition we have employed, but we are reluctant to repeat the analysis using substantially different definitions of the SSW such as the moment analysis employed by some recent studies because this would be a large amount of work; Butler et al. 2015 suggest that SSW rates are relatively robust to event definition in reanalyses. Extending the analysis to employ moment analysis and perhaps investigating in terms of split / displaced vortex SSWs would be very interesting, but outside the scope of this current study.

*10. L213-214; Restate the definition of the ENSO3.4 index here.*

Added definition on line 249.

*11. L214-216; Why is this metric chosen as a proxy for the Aleutian Low, instead of something simpler like the central pressure as in Overland et al. (1999)? Reference?*

The EOF method is from Chen et al. 2020 (a reference for which we have now added) who inspect the 1st EOF loading pattern of the North Pacific SLP then take a box average over a region where this pattern maximises. We use this measure as opposed to a fixed box method to allow for the possibility that the centre of the Aleutian Low in the model does not line up with observations. Taking the PC as opposed to a box over a maximum region in the EOF field seemed a cleaner method but below we check its robustness. We have amended the method with some better explanation of our metric (lines 250-255).

(Chen et al. 2020 show the SLP variability over the Aleutian islands is indeed the dominant EOF which explains 30% of total SLP variance in the northern pacific region. We have further analysed our metric and find a higher proportion of variance explained by the 1st EOF (38%) and that the 1st PC timeseries is highly correlated (r = 0.95) with a box area average over the maxima of the 1st EOF).

*12. L228-229; In my view, there looks to be a statistically significant difference in the number of SSW events distributed per month in Figure 1.*

This is true; we have amended the text acknowledging this bias in mid winter SSW rates but also outline that this type of bias in a model is relatively common and may originate in the discrepancy between dataset lengths (following analysis of *Horan and Reichler 2017),* We have also added a discussion of November SSWs, in response to comments from reviewer 3 (lines 270-278).

*13. L230-240; Although a comparison between model and reanalysis is great, it should still be noted that the samples are not completely comparable if SSWs are influenced by external forcing (climate change) in the real world.*

We have added a caveat stating this (lines 274-278). We also mention that the nature of climate change signal in SSWs is not well understood as outlined in Ayarzagüena et al. 2020.

*14. L239-246; Again, additional caveats about the use of one model for this analysis are needed... i.e., difference in QBO period (common in high-top models), which could affect the overall conclusions.*

We have added a caveat and acknowledgement of possible influence of these biases (lines 540-545)

*15. L321-322; This 90-year periodicity looks somewhat large though in Figure 8? Is there something physically-related to this or is it internal noise? Have you done any lead-lag or regression composites to further investigate any ENSO-SSW relationship at this time-scale?*

Apologies, not all the spectra used the same color scale and our discussion focussed more on interpretation of the significance contours and not the power values. We have re-plotted all spectra figures to show the same shading levels. When this is done one can see that ENSO exhibits significantly weaker power at relevant periods than the QBO metric. It is also worth pointing out that each spectra is normalised by the variance of the time series (this was probably not clear from our existing text and we have also rectified this in section 2.3).

We have also included new analysis using multilinear regression analysis, comparing the contributions of ENSO, the AL and the deep QBO amplitude (tables 1-3) to SSW_5yr however we stress that, while results from this approach are easy to interpret, they do not directly tackle the problem posed here for two main reasons:

- The signals observed on the wavelet spectra are non-stationary (only persistent for ~450 years of the simulation) and therefore a regression analysis of the entire series may not fully reveal relevant signals.

- Regression analysis considers variability in time series on all timescales whereas we focus on 60-90 year variability when examining our cross spectra. This could be overcome by filtering the timeseries and this is included in analysis and mentioned briefly.

The regression analysis has quantitatively verified the results indicated by wavelet spectra and has strengthened our interpretation – we thank the reviewer for this suggestion.

*16. L360; It could also be just noise in the short reanalysis record.*

This may be the case, however both Lu et al. papers present a relatively compelling statistical case for fluctuations in HT strength as well as evidence for a physical explanation for the variability. We have added some text to note the issues of extracting signal from noise in such a short data sample on line 274.

*17. L366-367; This is difficult to see in Figure 10. Could the left six panels be modified Slightly?*

The plots have been modified to make the lines thinner so that the amplitude modulation is more obvious (and also figure 11 for consistency).

*18. L375-378; This conclusion seems particularly sensitive to the QBO definition. Any Thoughts?*

Yes this is a fair assessment. We want to be transparent in our choice of QBO measure so have demonstrated the differences in results when different QBO metrics are used. All definitions show power at the 2-4 year periods (figure 10) but the longer-term variability is much noisier and more sensitive to the QBO level employed. Some of this may simply be noise (especially the shorter-lived responses) but some are likely to be real sporadic connections, further emphasising the presence of non-stationarity in the signal. It may be that the HT relationship is sensitive to different QBO levels at different times, perhaps depending on the strength of the planetary wave forcing from the troposphere, or on the predominance of wave 1 or wave 2 forcing, which could vary over time in some way. However, the presence of an extended response at ~60-90 years only in the 20 hPa and the deep QBO index plots supports our suggestion that the amplitude modulation of the westerly phase of the deep QBO (which can be seen by eye in fig 10a-f) is important at these timescales.

*19. L409-412; Where is this shown?*

We have not shown this explicitly in a figure - it was calculated from the ERA-Interim data shown in figures 2 and 3.

*20. L416-418; Reword sentence to improve clarity.*

We have reworded this to "In particular, the deep QBO index exhibited significant signals coincident with those in SSW_5yr corresponding to periodicities of around 90 years." (now found on line 533)

***Technical Comments:***
*1. L6; ". . . coupled Atmosphere-Ocean-Land-Sea ice model." to "coupled global climate model."*

Changed

*2. L46; ". . .and the [stratospheric polar] vortex."*

Added in stratospheric polar

*3. L49; "link" to "effect"*

*Changed*

*4. L74 and throughout; Unless you are talking about the vertical structure of the Aleutian Low, change "depth" to something like "strength" or "intensity"*
Changed to intensity throughout.

*5. L91; Lowercase "tropical"*
Changed to tropical

---

## Author Response (AR2)

**Response to Reviewer Comments**

**"Origins of Multi-decadal Variability in Sudden Stratospheric Warmings" by Oscar Dimdore-Miles et al.**

We thank all the reviewers for providing a second set of comments on our analysis. Their Suggestions have helped us clear up technical points as well as discuss more clearly the lower-than-expected covariation between the Aleutian Low and the vortex which is recommended for further study. Below is a summary of the relevant changes made to the manuscript.

**Reviewer 1**

The authors satisfactorily addressed the comments from my previous review, and the paper is nearly ready for acceptance. There are just a few remaining issues to clarify.

Major comments:
1. Both another reviewer and I were a bit confused by the relatively weak relationship between the AL index the authors chose and the SSW index. The authors, in their response to reviewer 3, note that if the box is pushed further north to match the region of Garfinkel et al 2010, the correlation is even lower. I find this surprising. Supplemental Figure 6 in Schwartz and Garfinkel 2020 shows the lag correlation between 500hPa height and wave1+2 heat flux at 100hPa in the UKMO model included in the S2S database (which I assume is very similar to the configuration used in this paper), and find that a box further north is indeed more closely associated with heat flux in the lower stratosphere than what the authors include.

I think a full investigation is beyond the scope of this paper, but I wonder if the box further north is more correlated with the seasonal mean vortex, while the definition used in this paper is (apparently) more closely associated with SSW. Even if a full discussion is deferred for future work, I think this issue merits some discussion in the paper itself.

This is a good point to raise and we agree that the nature of the connection between the AL and the vortex is not fully understood in this model. We have amended the results section (L437-442) reporting the lower correlation between the AL and SSWs when we utilise the box recommended in Garfinkel et al. 2010 and added comments in the discussion section (L532-537) alluding to the need for further study to diagnose the potential bias in the model teleconnection.

Minor comments,
Line 106: A more appropriate paper to cite than Garfinkel et al 2010 is Garfinkel et al 2020 (Climate Dynamics).
This citation has been changed (L106).

Line 542-549 and 567: Rao et al 2020 document the Holton Tan effect in the historical simulations of this model, and find that the model has a too-weak HT effect if a simple composite based on lower stratospheric winds is created (similar to most models), but does a better job if care is chosen to focus on the QBO phase that is most closely associated with

the HT effect. This could be mentioned in the discussion when you mention the caveat that this is just one model, as this model does a better job than most for the HT effect.
Reference to this work in the context of QBO and HT representation of UKESM and its impact on the findings on this study (L556).

Figure 10 and 11 are missing colorbars. If the colorbar is the same as other figures, please just note this fact.
This has been rectified with a note that the colour scale is the same as all other wavelet spectra figures.

**Reviewer 2**
General Comments:
Overall, this is a substantially improved manuscript and includes a better discussion on the physical connections between SSW variability and possible internal/external forcings. I thank the authors for their careful attention in replying to all of the Reviewers' concerns. I only have a few (very minor) technical comments that are listed below.

Technical Comments:
1. L128; Remove the word "unexpected"
This has been removed (L128).
2. L255; So not November as stated in L240?
We have clarified that while we do evaluate the warmings over the whole winter (Nov-Mar), we only use this to compare the SSW rate over the whole season with ERA-Interim. For the wavelet analysis and results studying coverability with other parts of the climate system, we use Dec-Mar warmings (L244-246).
L347; "…shown on the right of figure 6…"
This has been added (L349).
3. L381-384; In the real world there could also be an influence of volcanic aerosols (natural variability) in the stratosphere. But I guess this is not the case in the model, as stated in L145?
This is correct, the volcanic aerosol forcing in the simulation is a constant, prescribed, background contribution so could not account for variability found.
4. L550; Restate the model name here.
This has been added (L563).
5. L567-580; I am not sure all of this context is necessary in the conclusions, as I found it a bit distracting and taking away from the main contributions of your work.
We have removed much of this context to make our contribution clearer in the closing sentences (L579-587).

**Reviewer 3**
The authors have addressed most of my comments and suggestions. I appreciate the efforts to improve the description of the wavelet technique and to extend the discussion of the potential role of the PDO in defining the multi-decadal variability of SSWs. I have just a couple of small comments, mostly regarding new text included during the review process.

L255: the authors indicate that the month range considered in the Aleutian low index is the same as that used for their SSWs definition. However, the months used for the SSWs definition is stated some sentences later.

This is a good point and the order of introducing the month range over which we evaluate the AL and the SSWs has now been rectified (L244-246).

L282: I would not recommend describing this model bias as cold pole bias. The cold pole bias traditionally refers to the bias of many models towards an anomalously strong climatological vortex (particularly in the Southern Hemisphere). Usually this is due to a lack of wave forcing in models. In the present study, it does not seem that the model simulates a too strong vortex, since the averaged frequency of SSWs is not that different from that in observations. Instead, the vortex is probably too weak in November and so there is an unrealistic frequency of SSWs in that month. As the vortex is recovering in the following months, the occurrence of an SSW becomes less likely. Nevertheless, it is true that it is a common problem across models as shown by Charlton et al. (2007) or more recently by Ayarzagüena et al. (2020).

We have removed reference to a cold polar bias (L275).

---

## Author Response (AR3)

**Response to Editor Corrections**

**"Origins of Multi-decadal Variability in Sudden Stratospheric Warmings" by Oscar Dimdore-Miles et al.**

We thank the editor for their final corrections to the manuscript. We have addressed all points raised and added the recommended references. We thank the Editor for their help over the whole submission process as well as other members of the team at WCD. We look forward to submitting other work to your journal in the future.

Kind Regards

The Authors.